# FULLY DIFFERENTIABLE METRIC TEMPORAL FIRST-ORDER RULE LEARNING

## ABSTRACT

We propose a novel differentiable neural architecture for learning first-order temporal logic rules enriched with metric operators. Leveraging differentiable immediate consequence operators over data, we extend the approach to temporal data by learning both the predicates and the temporal intervals in which they hold. Among the strengths of our model are its support of existential literals in rule bodies to express eventualities within an interval and its applicability to data over discrete intervals. Notably, our model can effectively capture temporal dependencies without reifying all possible timestamps and produces a linear number of rules in the size of the training set, which has a benign effect on model complexity and scalability. We explore different use cases and show in experiments the benefits of our approach, highlighting its potential as a scalable solution for interpretable metric temporal rules over data.

## 1 INTRODUCTION

Inductive Logic Programming (ILP) (Muggleton, 1991; Cropper & Dumancic, 2022) is a field that seeks to derive knowledge expressed as logic programs from training examples, enabling generalization to unseen cases. In recent years, the interest in ILP has been increasing since symbolic approaches grounded in logic rule learning are by their nature well-positioned for providing explainability, which is a pressing need for modern data-driven AI systems.

Specifically, with the growing availability of temporal data, learning temporal rules from such data has become an issue in the research community. Often temporal data are relational data enriched with temporal annotations, representing dynamic events as tuples of the form $(s, r, o, t)$ or $(s, r, o, [t_1, t_2])$, where $t$ is a timestamp and $[t_1, t_2]$ a time interval associated with a data triple $(s, r, o)$. Such representations are ubiquitous and critical for domains like international diplomacy (Boschee & Lautenschlager 2015), healthcare (Chaturvedi 2024), and finance (Jeyaraman et al. 2024), where the temporal validity of facts influences decision-making.

Logic rules can express time in different ways, including (i) the use of designated arguments and functions to encode successive states as in (Chomicki & Imielinski, 1988; Eiter & Šimkus, 2010; Ronca et al., 2018) for linear structures or as in (Chomicki & Imielinski, 1993; Eiter & Simkus, 2009) for branching time, (ii) expressing temporal constraints (Janhunen et al., 2017) (iii) molding action domains (Gelfond & Lifschitz, 1998; Eiter et al., 2005) or (iv) directly by supporting temporal modalities, e.g. (Aguado et al. 2023; Beck et al. 2018; Wałęga et al. 2021).

Recent works (Liu et al., 2022; Xiong et al., 2024; Wang et al., 2024) extended logical rule learning to temporal settings but are limited in expressiveness, typically restricting reasoning to point-based timestamps, simple forward-propagating patterns, or disregarding temporal uncertainty, i.e., when events happen. In this work, we address these limitations by proposing a novel framework for learning *metric temporal rules* Walega et al. (2019). The latter are a symbolic formalism that extends Datalog with temporal operators such as "always" ($\square$), "historically" ($\blacksquare$), "eventually" ($\lozenge$), and "once" ($\blacklozenge$) from linear time logic (LTL), each annotated with explicit time intervals. For example, the rule

$$sign\_fa(Y, X) \leftarrow \blacklozenge_{[2,3]} eoi(X, Y), \blacksquare_{[0,2]} visit(Y, X), \square_{[0,2]} visit(Y, X).$$

says that $Y$ may sign a formal agreement with $X$ if the latter expressed an intent to meet two or three days ago, and visits of $Y$ to $X$ will continue for two days from now since two days ago. Such rules form a core fragment of DatalogMTL (Wałęga et al. 2021) and LARS (Beck et al. 2018).

To achieve this, we introduce the first fully differentiable architecture for metric temporal rule learning, see Figure 1. Leveraging neural networks, our approach MT-Diff-Learn learns symbolic rules in an end-to-end manner while retaining the same semantics as the one adopted by the standard metric temporal reasoners already available (Wang et al. 2022; Beck et al. 2017). In contrast to embedding-based approaches, where reasoning is performed on latent representations in a vector space that encodes data and knowledge, e.g. (García-Durán et al., 2018) (see Section 5), our method produces human-readable rules that can be applied for downstream reasoning or analysis.

Our main contributions are summarized as follows:

- We introduce a new mechanism for learning eventuality conditions $\Diamond$ and $\blacklozenge$ within temporal intervals, allowing the model to capture rules that require relevant events to occur *somewhere* within a specified interval (as in the example rule above). This substantially increases the expressiveness of learned temporal rules and extends the approach in (Wang et al. 2024), which only supports as "always" where only *everywhere* operators $\Box$ and $\blacksquare$ within a specified interval.
- By its design, the framework is the first offering fully differentiable metric rule learning in an end-to-end manner using neural networks. This paves the way for integrating temporal reasoning into larger differentiable architectures and enables seamless gradient-based training.
- We demonstrate the applicability of our approach for three use cases, showing that it is capable of producing not only more succinct (and thus intuitively more general and easier to read) temporal rules than other methods, but may also increase performance.

## 2 PRELIMINARIES

In this section, we first present DatalogMTL, followed by the differentiable immediate consequence operator. This will lay the groundwork for introducing our differentiable immediate consequence operator for metric temporal rules in the next section.

**DatalogMTL** We consider a fragment of DatalogMTL (Wałęga et al. 2021), which extends Datalog with metric operators, where LTL operators such as "always" $\Box$, "historically" $\blacksquare$, "eventually" $\Diamond$, "once" $\blacklozenge$ (Koymans, 1990) are annotated with intervals. It builds on *metric atoms* of the form

$$M ::= \blacksquare_\rho P(\mathbf{s}) \mid \Box_\rho P(\mathbf{s}) \mid \blacklozenge_\rho P(\mathbf{s}) \mid \Diamond_\rho P(\mathbf{s})$$

where $P(\mathbf{s})$ is a relation atom, i.e., $\mathbf{s}$ is a tuple of variables and constants with the same arity as $P$, and $\rho$ is an interval of non-negative numbers; we also use $P(\mathbf{s})$ as a shorthand for $\Box_{[0,0]} P(\mathbf{s})$ (the singleton interval referring to the current time point. E.g., $\blacklozenge_{[2,3]} eoi(X, Y)$ holds at time $t$ if $X$ expressed an intent to meet $Y$ in the interval between two or three days ago. A *(metric) rule* is an expression of the form

$$P(\mathbf{s}) \leftarrow M_1, \ldots, M_n \quad \text{for } n \geq 1 \tag{1}$$

where the *body atoms* $M_1, \ldots, M_n$ are metric atoms and the *head atom* $P(\mathbf{s})$ is relational. A *program* is a finite set of rules. A *temporal dataset* is a finite set of temporal facts $P(\mathbf{c})@t$ where $P(\mathbf{c})$ is a ground (i.e., variable-free) relational atom and $t \in \mathbb{Z}$.

An *interpretation* $I$ is a function assigning a truth value (0 or 1) to each ground relational atom $P(\mathbf{c})$ and time point $t \in \mathbb{Z}$. The satisfaction of relational and metric atoms, is inductively defined:

- $I, t \models P(\mathbf{c})$ if $I(P(\mathbf{c}), t) = 1$, otherwise $I, t \not\models P(\mathbf{c})$.
- $I, t \models \blacksquare_{[a,b]} P(\mathbf{c})$ if for all $t' \in [t - b, t - a]$, $I, t' \models P(\mathbf{c})$.
- $I, t \models \Box_{[a,b]} P(\mathbf{c})$ if for all $t' \in [t + a, t + b]$, $I, t' \models P(\mathbf{c})$.
- $I, t \models \blacklozenge_{[a,b]} P(\mathbf{c})$ if some $t' \in [t - b, t - a]$ exists such that $I, t' \models P(\mathbf{c})$.
- $I, t \models \Diamond_{[a,b]} P(\mathbf{c})$ if some $t' \in [t + a, t + b]$ exists such that $I, t' \models P(\mathbf{c})$.

We say that $I$ satisfies a rule $r$ of form (1) at time $t$, denoted $I, t \models r$, if either $I, t \not\models M_i$ for some $i \in \{1, \ldots, n\}$ or $I, t \models P(\mathbf{s})$. Furthermore, $I$ satisfies a program $\pi$ at time $t$, denoted, $I, t \models \pi$, if $I, t \models r$ for each $r \in \pi$. Finally, $I$ is a model of $\pi$, denoted $I \models \pi$, if $I, t \models \pi$ for every $t \in \mathbb{Z}$.

We adopt *Herbrand interpretations*, substituting each variable with all constants from the rules or dataset $D$, and denote this domain by $\mathcal{U}$. We assume $\mathcal{U}$ is nonempty, as reasonable datasets contain at least one element (otherwise, one can be added). For a program $\pi$ and a domain $\mathcal{U}$, $\pi_g[\mathcal{U}]$ is the *grounded version* of $\pi$, obtained by substituting all variables with all possible combinations of constants appearing in $D$. While $\pi_g[\mathcal{U}]$ may admit multiple models, Wałęga et al. (2021) showed it has a single *minimal model*, which is computable by repeated applications of *materialisation-based reasoning algorithms*. The latter syntactically applies the rules of $\pi_g[\mathcal{U}]$ over a dataset $D$ to simulate the behavior of the *immediate consequence operator*.

Formally, for a rule $r$ of the form (1) the set $T_r[D]$ consists of all temporal facts $P(\mathbf{s})\nu$ at $t$ such that:

- $\nu$ is a substitution replacing all variables in $r$ with constants from $D$ (i.e., $\nu$ grounds $r$), and
- $D, t \models M_i\nu$ for all $1 \le i \le n$, viewing $D$ as interpretation (i.e., the grounded body of $r$ holds at time $t$ in $D$).

We set $T_\pi[D] = \bigcup_{r \in \pi} T_r[D]$. Iterating from $D_0 = D$, the minimal model of $\pi_g[\mathcal{U}]$ is obtained as the least fixed point, denoted $\mathrm{lfp}_\pi[D]$.

**Differentiable Immediate Consequence for ILP** In *Inductive Logic Programming* (ILP) (Muggleton et al., 2012; Cropper & Dumancic, 2022), the objective is to learn a logic program $P$ that can derive a designated *target atom* $h$, given background knowledge $B$ and sets $\mathcal{P}$ and $\mathcal{N}$ of positive examples and negative examples, respectively. A solution $P$ must entail all positive examples while excluding all negative ones: $B \cup P \models e^+$, $\forall e^+ \in \mathcal{P} \wedge B \cup P \not\models e^-$, $\forall e^- \in \mathcal{N}$.

In algebraic ILP frameworks, the canonical immediate consequence operator $T_\pi$ for first-order (atempoal logic programs) can be formulated using matrix operations combined with a threshold function (Gao et al., 2024), where propositional atoms are viewed as real-valued variables. Let $v$ be an real number, and let $\varphi(v) = v'$ be the threshold function, where $v' = 1$ if $v \ge \tau$, and $v'_i = 0$ otherwise with $\tau$ being the threshold value.

Given a logic program $\pi_h = \{r_1, \ldots r_m\}$ of rules that share the same predicate in the head $h$ and a substitution $\theta$, the algebraic immediate consequence operator $D_{\pi_h}$ computes the Boolean value of the grounded head atom for $\boldsymbol{v}$ as: $D_{\pi_h}(\boldsymbol{v}_h) = \bigvee_{k=1}^m \varphi(\mathbf{M}_h[k,\cdot]\,\boldsymbol{v}_h)$, where $\mathbf{M}_h[k,\cdot]$ denotes the $k$-th row of the program matrix $\mathbf{M}_h$ (Gao et al., 2024). The vector $\boldsymbol{v}_h$ encodes the Boolean values of the body atoms of $r_k$ under $\theta$, and $D_{\pi_h}(\boldsymbol{v}_h)$ computes whether the corresponding head atom $h$ is derivable under $\theta$. Note that for each first-order atom $h$, a different matrix $\mathbf{M}_h$ is considered.

To enable gradient-based learning, Gao et al. (2024) proposed a differentiable approximation of $D_{\pi_h}$. Specifically, a trainable matrix $\tilde{\mathbf{M}}_h$ with values from the interval $[0, 1]$ represents the learnable rule weights. The discrete threshold function $\varphi$ is replaced by the differentiable sigmoid activation: $\Phi(x) = \frac{1}{1+e^{-\eta x}}$, where $\eta$ controls the smoothness of the approximation. Disjunction is replaced in $D_{\pi_h}$ by a *fuzzy disjunction layer*: $\mathcal{F}(\boldsymbol{x}) = 1 - \prod_{i=1}^n (1-x_i)$, which is a differentiable approximation.

The forward computation of the approximation of $D_{\pi_h}$ is thus defined as: $\tilde{\boldsymbol{v}}_o = \mathcal{F}(\Phi(\tilde{\mathbf{M}}_h \boldsymbol{v}_h - \mathbf{1}))$, where $-\mathbf{1}$ is to align the sigmoid activation, which is applied to the vector component-wise, with the original threshold function. By minimizing the loss between $\tilde{\boldsymbol{v}}_o$ and the ground-truth interpretation vectors, the neural network learns $\tilde{\mathbf{M}}_h$ as a differentiable representation of rules with head $h$. The collection of all $D_{\pi_h}$ is denoted by $D_\pi$.

## 3 MT-DIFF-LEARN

We build upon the approach by Gao et al. (2024), who designed a neural network to mimic $D_\pi$ during forward computation on knowledge graphs. We extend it by learning rules with metric operators in the body, enabling the learning of both the predicates and the specific intervals in which they hold. In the sequel, we present the constituents of the approach, which are data preparation, construction of the neural network, and rules extraction, see Figure 1.

**Data Preparation** The preprocessing pipeline takes as input a set of temporal facts and, using a sliding window-based approach, generates first-order atoms such as $visit(X, Y)$. They represent different ground atoms, e.g., $visit(Angela\_Merkel, Italy)$ or $visit(Barack\_Obama, Germany)$

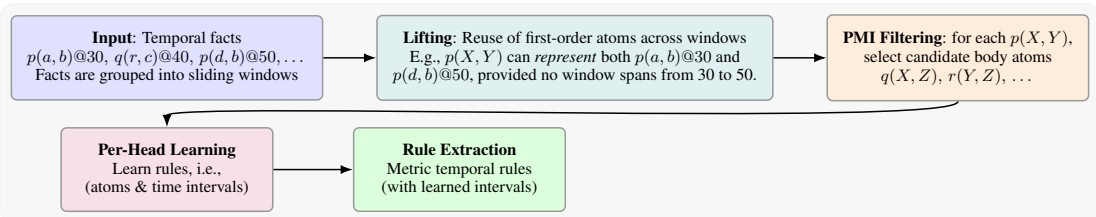

Figure 1: Overview of the MT-Diff-Learn pipeline.

in different windows. We *lift* ground atoms to first-order atoms to obtain more general rules that can be instantiated in multiple ways. To ensure soundness, each variable $X$ refers to a single entity within a window, but may be *reused* across different windows to promote generalization.

---

**Algorithm 1** Windowing, Window Ordering, and Lifting to First-Order Atoms

---

1: **Input:** Temporal facts $D$, window size $l$.
2: **Output:** A dictionary mapping each first-order atom $p(X_i, X_j)$ and window id $w$ to the grounded atoms it represents $p(e_i, e_j)$.
3: **(1) Windowing**
4: Partition timestamps into windows $\mathcal{W} = \{w : [w, w + l] \mid w \in [0, |t_{\max} - l|]\}$.
5: For each window $w \in \mathcal{W}$, collect all facts with timestamps in $[w, w + l)$ and merge consecutive $p(e_i, e_j)$ occurrences into centered intervals.
6: **(2) Window ordering**
7: For each $w \in \mathcal{W}$, compute the overlap score $O(w)$ as the number of entity pairs shared with any other window.
8: Sort windows by decreasing $O(w)$; break ties by number of facts then index.
9: **(3) Lifting ground atoms to first-order atoms**
10: **for** each window $w$ in sorted order **do**
11:     Initialize $\sigma_w : E_w \to V = v_0, v_1, \ldots$ where $E_w$ denotes entities appearing win window $w$.
12:     Sort all entity pairs in $w$ by their global frequency (highest first).
13:     **for** each entity pair $(e_i, e_j)$ in this order **do**
14:         Select any ground atom $p(e_i, e_j)$ from $w$;
15:         If a first-order atom $p(X_i, X_j)$ was generated in previous windows and $X_i, X_j$ are not yet assigned in $w$, set $\sigma_w(e_i) = X_i$ and $\sigma_w(e_j) = X_j$;
16:         else if either $e_i$ or $e_j$ is already mapped in $w$, reuse the mapped variable and introduce a fresh one if needed to add $p(X_i, X_k)$ or $p(X_k, X_j)$;
17:         else introduce fresh variables $X_m$ and $X_n$ and add $p(X_m, X_n)$;
18:         Record the resulting first-order representative $p(X_i, X_j)$;
19:     **end for**
20: **end for**

---

In real-world datasets, the first-order atoms produced by the procedure above may correspond to ground atoms that co-occur with many others across different windows. Such co-occurrences can be highly informative for making inferences, as they capture meaningful temporal and relational patterns. However, in practice the number of potential co-occurring pairs is often extremely large, making it necessary to distinguish informative associations from spurious or incidental ones. To this end, we compute the Pointwise Mutual Information (PMI) Church & Hanks (1990) between atoms, which quantifies the statistical dependence between two predicates:

$$\text{PMI}(a, b) = \log \frac{P(a, b)}{P(a)P(b)}$$

where $P(a, b) = C(a, b)/|\mathcal{W}|$ represents the frequency that atoms $a$ and $b$ appear together in a window as a ratio between the number of co-occurrences $C(a, b)$ and the number of windows $|\mathcal{W}|$, and $P(a) = \sum_{w \in \mathcal{W}} \mathbf{1}_{\{a \text{ appears in } w\}}/|\mathcal{W}|$ denotes the frequency that atom $a$ appears in a window (and analogously for $P(b)$). In the definition of PMI, if the denominator equals zero, then the value is $-\infty$. Intuitively, PMI measures how much more often two atoms co-occur than would be expected if they were statistically independent.

Given a target (head) atom $h = p(X, Y)$, we select candidate body atoms by exploring a PMI-weighted co-occurrence graph using a bounded breadth & depth search controlled by two parameters: (i) the *breadth* $b$ which limits the number of atoms expanded at each layer and (ii) the *depth* $d$. The following procedure conducts a constrained breadth-first exploration: $b$ and $d$ bounds the total exploration depth.

---

**Algorithm 2** PMI Computation and Candidate-Body Selection

---

1: **Input:** First-order atoms $\mathcal{A}$, windows $\mathcal{W}$, breadth $b$, depth $d$.
2: **Output:** Candidate body atoms for each head atom.
3: **(1) PMI computation**
4: **for** each pair of atoms $(a, b)$ **do**
5:     Count co-occurrences $C(a, b)$ across windows.
6:     Compute $P(a)$, $P(b)$, $P(a, b)$.
7:     Compute $\mathrm{PMI}(a, b) = \log\big(P(a, b)/(P(a)P(b))\big)$.
8: **end for**
9: **(2) Candidate-body selection**
10: **for** each head atom $h$ **do**
11:     *// Layer 0: per-head-variable selection*
12:     For each variable $X$ in $h$, pick up to $b$ atoms sharing $X$ with $h$ and with highest $\mathrm{PMI}(h, \cdot)$, and add them to the candidate pool;
13:     *// Layers 1..d: PMI-based neighborhood expansion*
14:     For each level $1..d$, expand every atom in the current layer by adding up to $b$ new neighbors sharing a variable with it; the newly added atoms form the next layer.
15: **end for**

---

Our preprocessing method is heuristic but grounded in well-established statistical principles such as PMI. Crucially, MT-Diff-Learn does not depend on these specific design choices: the subsequent components—and in particular the neural layer implementing the differentiable immediate consequence operator for metric rules—operate unchanged under any alternative procedure that produces first-order atoms and candidate body predicates. In this sense, the preprocessing stage is modular and can be replaced or refined independently without affecting the core architecture.

**Neural Module: Differentiable Metric Immediate-Consequence Operator** For each target atom $h = p(X, Y)$, the neural operator receives, for every relevant atom $\gamma_j$, information about whether it holds in a window $w$ and, if so, the intervals during which it holds. This information is conveniently represented by three input tensors: the *truth tensor*, the *start tensor*, and the *end tensor*. These tensors are constructed after the PMI-based pre-filtering step, which selects $n^h$ relevant atoms $\gamma_0, \ldots, \gamma_{n^h-1}$ for the target atom (see previous section); note that by construction $n^h \le b^{d+1}$. More in detail, for each target atom $h$ we want to derive, we have the following input:

- a *truth tensor* $\mathbf{T}^{\mathrm{in}}$ of shape $(|\mathcal{W}|, n^h)$, where each entry is 1 if the relevant atom $\gamma_j$ is active in window $w$ and 0 otherwise. This corresponds to the matrix $\mathbf{M}_h$ introduced in the preliminaries;
- a *start tensor* $\mathbf{S}^{\mathrm{in}} \in \mathbb{R}^{|\mathcal{W}| \times n^h \times \mathrm{max\_int}}$, where $\mathbf{S}^{\mathrm{in}}_{w,j,k}$ stores the starting point of the $k$-th interval in which $\gamma_j$ holds in window $w$. Multiple interval slots are needed because a relevant atom may hold in several disjoint subintervals.
- an *end tensor* $\mathbf{E}^{\mathrm{in}}$, which, instead, represents the ending points of those intervals.

If a relevant atom occurs in fewer than max_int disjoint intervals, the remaining slots are filled by repeating its last start and end values; if it does not occur in a window at all, we instead assign $+\infty$ and $-\infty$ as the default start and end values, respectively.

In our approach, we limit the range of the learnable intervals to the upper-bound implied by the considered window size. An important note is that in approaches involving a translation into a symbolic inductive logic learner, e.g. the one in (Wang et al. 2024), the input to the rule learner increases *quadratically*, as in principle one needs to reify each candidate body atom per possible timestamp. Our approach needs no reification as we consider intervals as first-class citizens; we learn them by learning the lower and upper bounds as numbers, as explained below.

For each target atom $h$, the model learns a matrix $\mathbf{T} \in [0, 1]^{n^h_r \times 3n^h}$, where $n^h_r$ is the number of rules associated with $h$ (equal to the number of grounded atoms represented by $h$). Each row of

$\mathbf{T}$ sums to 1 over $3n^h$ entries and is divided into three segments of equal size: (i) the first third selects always-metric literals with an associated proper interval; (ii) the second third selects punctual literals (holding at a single timestamp); (iii) the final third selects eventuality-metric literals with an associated proper interval. The value $\mathbf{T}_{i,j}$ specifies the degree to which the $j$-th relevant atom $\gamma_j$ is included in the body of rule $r_i$. A threshold $\tau$ is later used to determine whether $\gamma_j \in [0,1]$ is retained as an actual body literal.

In addition, the model learns matrices $\mathbf{S}^A, \mathbf{E}^A, \mathbf{P}, \mathbf{S}^E$, and $\mathbf{E}^E \in \mathbb{R}^{n_r^h \times n^h}$. For an always-metric literal, $\lfloor \mathbf{S}_{i,j}^A \rfloor$ and $\lceil \mathbf{E}_{i,j}^A \rceil$ define the learned lower and upper bounds of its interval; for an eventuality-metric literal, the corresponding bounds are given by $\lfloor \mathbf{S}_{i,j}^E \rfloor$ and $\lceil \mathbf{E}_{i,j}^E \rceil$. The matrix $\mathbf{P}$ contains the learned timestamps for punctual literals.

The forward pass implements a differentiable approximation of the metric immediate-consequence operator. Conceptually, for each window $w$ and each target atom $h$, the network performs the following computational steps.

**Step 1: Normalize rule–body weights.** Each row of the matrix $\mathbf{T}$ is normalized via $\tilde{\mathbf{T}} = \text{Softmax}(\mathbf{T}, -1)$, ensuring that, the weights assigned to the three temporal variants (always, punctual, eventually) of each relevant atoms $\gamma_j$ form a probability distribution (i.e., rows sum to 1).

**Step 2: Compute "always" activations (interval containment).** For each learned interval $[\mathbf{S}_{i,j}^A, \mathbf{E}_{i,j}^A]$, the operator computes differentiable containment scores:

$$\mathbf{A}^{[} = \sigma(\mathbf{S}^A \otimes \mathbf{1} - \mathbf{S}_{w,:}^{\text{in}}), \qquad \mathbf{A}^{]} = \sigma(\mathbf{E}_{w,:}^{\text{in}} - \mathbf{E}^A \otimes \mathbf{1})$$

where broadcasting via $\otimes \mathbf{1}$ aligns learned bounds with the max_int ground-truth intervals. Containment over all possible interval indices $k$ is aggregated by:

$$\mathbf{A}^A = \mathcal{F}(\mathbf{A}^{[} \odot \mathbf{A}^{]}, -1).$$

**Step 3: Compute punctual activations (singleton intervals).** To process degenerate intervals:

$$\mathbf{A}^P = \mathcal{F}(\sigma(\mathbf{P} \otimes \mathbf{1} - \mathbf{S}_{w,:}^{\text{in}}) \odot \sigma(\mathbf{E}_{w,:}^{\text{in}} + 1 - \mathbf{P} \otimes \mathbf{1}), -1).$$

The $+1$ term enables a containment check for *degenerate* intervals: if the model learns a punctual interval at time $t$, we can test containment by verifying whether a candidate point $t'$ satisfies $t' \in [t, t+1]$. During rule extraction, this effect is reverted by taking the floor of the learned value $t'$, which correctly yields the intended punctual (degenerate) interval, namely $[\lfloor t' \rfloor, \lfloor t' \rfloor]$.

**Step 4: Compute "eventually" activations (interval overlap).** Eventuality conditions require that the learned interval for $\gamma_j$ overlaps *at least once* with one of the ground-truth intervals in window $w$. This is enforced through two activation matrices: (i) $\mathbf{A}^{\nleq} = \sigma(\mathbf{E}^{\text{in}} - \mathbf{S}_{w,:}^E \otimes \mathbf{1})$, ensuring that the learned interval starts before the ground-truth one ends; and (ii) $\mathbf{A}^{\ngeq} = \sigma(\mathbf{E}_{w,:}^E \otimes \mathbf{1} - \mathbf{S}^{\text{in}})$, ensuring that the learned interval ends after the ground-truth one begins. These correspond, respectively, to the necessary and sufficient conditions for an overlap in a proper interval, the case of the degenerate interval is covered by the punctual activation (Step 3). The fuzzy conjunction is then applied:

$$\mathbf{A}^E = \mathcal{F}(\mathbf{A}^{\nleq} \odot \mathbf{A}^{\ngeq}, -1).$$

**Step 5: Compute body activations.** The final body activation vector concatenates the contributions of the three temporal forms:

$$\mathbf{B} = \tilde{\mathbf{T}}[:, 0:n^h] \odot \mathbf{A}^A \odot \mathbf{T}_{w,:}^{\text{in}} \,\Big\|\, \tilde{\mathbf{T}}[:, n^h:2n^h] \odot \mathbf{A}^P \odot \mathbf{T}_{w,:}^{\text{in}} \,\Big\|\, \tilde{\mathbf{T}}[:, 2n^h:3n^h] \odot \mathbf{A}^E \odot \mathbf{T}_{w,:}^{\text{in}},$$

where $\|$ denotes concatenation. Therefore the shape of $\mathbf{B}$ is $(n_r^h, 3n^h)$ for each target atom $h$.

**Step 6: Apply structural penalties.** To encourage (i) *diversity among bodies* and (ii) *presence of head variables in body atom variables*, the model incorporates, respectively, a cosine similarity penalty and a masked-based penalty. The pairwise cosine similarity matrix is computed as $\text{CosSim}_{i,j} = \widehat{\tilde{\mathbf{T}}}_i \cdot \widehat{\tilde{\mathbf{T}}}_j$ for $i \neq j$, where $\widehat{\tilde{\mathbf{T}}}_i = \tilde{\mathbf{T}}_i / \|\tilde{\mathbf{T}}_i\|_2$ measures similarity between rules from $\ell_2$-normalized $\tilde{\mathbf{T}}$, where $i \in \{1, \dots, n_r^h\}$. To avoid self-comparison, diagonal values are ignored

(set to 0). Let $\text{AvgSim}_i = \frac{1}{n_r^h - 1} \sum_{j \neq i} \text{CosSim}_{i,j}$ be the average similarity of the $i$-th rule (row). To foster diversity in training, we use a decay factor $\alpha_i = 1 - \text{AvgSim}_i \cdot (1 - \text{epoch/max\_epoch})^2$ based on the current and the total number of epochs.

To guide the training toward producing first-order rules, we introduce a mask-based penalty as follows. We compute two *mask*-vectors $\mathbf{m}^0$ and $\mathbf{m}^1$ such that $\mathbf{m}_k^0$ is 1 if the $k$-th atom contains the first variable of the target atom, otherwise 0. The $\mathbf{m}^1$ similarly captures variable sharing with the second argument of the target atom. Let $w^k$ for $k \in \{0,1\}$ be defined as $w^k = \text{Sum}(\tilde{\mathbf{T}} \odot (\mathbf{1} \otimes (\mathbf{m}^k || \mathbf{m}^k || \mathbf{m}^k)), -1)$; $w^0$ (resp. $w^1$) is greater than *threshold* $\tau$ only if some body atom contains the same first (resp. second) variable as the head atom. We then compute a penalty

$$p^k = 1 - \text{Relu}(\tau - w^k) \cdot (1 - \text{epoch/max\_epoch})^2$$

The *threshold* parameter can be selected in a grid search, choosing the value yielding the best result.

**Step 7: Aggregate rule consequences.** The output of the operator is obtained through a fuzzy disjunction over rules:

$$O = \mathcal{F}\big(\Phi\big(p_i^0 \, p_i^1 \, \alpha_i \cdot \mathbf{B} \, \mathbf{1}_{3n^h}\big)\big),$$

where $\mathbf{1}$ is the all-ones vector and $\Phi$ the differentiable aggregation within each rule.

While standard regularization adds a weighted penalty to the loss function, our mechanism enforces a soft diversity constraint directly in the forward pass, which dynamically down-weights redundant rules by reducing their *inference impact*. Note that those weights are high during the first epochs and diminish till, at the last step, their impact is zero.

**Training & Rule Extraction** For each window $w$, the network produces predictions $\hat{y}_h^w = f_\Theta(w)$ for every target atom $h$, which are compared against ground-truth labels $y_h^w$. To account for the strong imbalance between positive and negative instances, we use a *balanced binary cross-entropy loss*. The *total loss* is the sum over all atoms, augmented with an $\ell_1$ regularization term on the parameters $\Theta$ to promote sparsity and reduce overfitting.

Once training is completed, we extract the learned rules by inspecting the learned weight matrices corresponding to each target atom. Specifically, for each target atom $h(X, Y)$, we consider the learned truth weights $\mathbf{T}$, start and end weights $\mathbf{S}^A, \mathbf{E}^A, \mathbf{S}^E$ and $\mathbf{E}^E$, respectively, and the punctual intervals $\mathbf{P}$. Each such matrix contains the same number of rows, which we read as the bodies of the rules having $h(X, Y)$ in the head. Let us consider $\tau$ as a threshold hyperparameter.

During the preparation of the data, we produced two variables that, for each window, represent the same entity. In this case, we choose one to be the *representative*, and we substitute all the occurrences of the other variables in the first-order atom signature. We categorize rules $r$ as follows:

- $r$ is *safe* if all head variables also appear in its body; e.g., $h(X, Y) \leftarrow \gamma_1(Y), \gamma_2(X, Z)$ is safe.
- $r$ is *partially safe* if at least one, but not all, of the head variables appear in the body. For instance, $h(X, Y) \leftarrow \gamma_2(X, Z)$ is partially safe.
- $r$ is *to ground* if no head variable appears in the body; e.g., $h(X, Y) \leftarrow \gamma_1(Z)$ is to ground.

We then proceed with rule $r$ as follows: if $r$ is safe, we keep the variables of each atom as they appear; if $r$ partially safe, we keep only the variables appearing in both its head and body. For each window in the training dataset, we ground the remaining variables with the entities they represent in that window; if $r$ is to ground, we proceed as in the previous case but all variables will be grounded.

Next, we describe how to extract the intervals for the body atoms $\gamma_j$ with $j = 0, \ldots, n^h - 1$, from the $i$-th rule $r$; recall that $i = 0, \ldots, n_r^h - 1$. For each included $\gamma_j = p(X, Y)$, we have two cases:

- **Metric (interval) case:** We extract as learned interval: $I_{\gamma_j} = [\lfloor \mathbf{S}_{i,j} \rfloor, \lceil \mathbf{E}_{i,j} \rceil]$, indicating the interval in the window where $\gamma_j$ is considered active; this allows the rule $r$ to capture temporal relations holding over a range of timestamps.
- **Punctual (single timestamp) case:** For $\gamma_i$ to hold at a specific timestamp, we extract the learned punctual time and respective interval for a rule $r$ as: $t_j = \lfloor \mathbf{P}_{i,j} \rfloor$ and $I_{\gamma_j} = [t_j, t_j]$.

We then represent $r$ as: $h(X, Y) \leftarrow \bigwedge_{\gamma_j \in B(r)} \bigwedge_{\text{op} \in \{\blacksquare, \square, \blacklozenge, \lozenge\}} \text{op}_{I_{\gamma_j}} p_{\gamma_j}(X_{\gamma_j}, Y_{\gamma_j})$ where $B(r)$ is the body of $r$, and $\text{op}_{I_{\gamma_j}} p_{\gamma_j}(X_{\gamma_j}, Y_{\gamma_j})$ indicates that the atom $\gamma_j$ holds over the learned interval, which is for interpolation is centered around the center of the window. For illustration, for window

size 8, the learned weights vary from 0 to 7 and the timestamp of the target atom is $\lfloor 8/2 \rfloor = 4$. Thus for $\gamma(X, Y)$, an always-learned interval $[0, 2]$ is transformed into $\blacksquare_{[2,4]}\gamma(X, Y)$, a learned interval $[2, 5]$ into $\blacksquare_{[0,2]}\gamma(X, Y) \wedge \square_{[0,1]}\gamma(X, Y)$, and a learned into $[6, 6]$ into $\square_{[2,2]}\gamma(X, Y)$.

For each rule $r$, let $s(r) = |\{H@t \mid D, t \models H, D, t \models B\}|$, $b(r) = |\{H@t \mid D, t \models B\}|$, and $h(r) = |\{H@t \mid D, t \models H\}|$, where $D, t \models B$ means $D, t$ satisfies the grounded body the atoms in $B$ represents in the window $w = t - \lfloor l/2 \rfloor$, with $l$ the window length. We set $\mathrm{sc}(r) = s(r)/b(r)$ if $b(r) > 0$ (else 0), and $\mathrm{hc}(r) = s(r)/h(r)$ if $h(r) > 0$ (else 0). The final weight is $w(h, r) = \beta\,\mathrm{sc}(r) + (1 - \beta)\,\mathrm{hc}(r)$ with $\beta \in [0, 1]$.

**Sliding Slice-Based Algorithm**   To address real-world temporal datasets and better capture localized temporal patterns, we propose a sliding window algorithm that partitions the dataset into overlapping slices. Formally, given a temporal dataset of facts $p(\bar{c}, t)$ where $\bar{c} = c_1, \ldots, c_n$, we define a sequence of temporal slices of length $l$ as: $\{p(\bar{c}, t) \mid 0 \le t < l\}, \{p(\bar{c}, t) \mid l \le t < 2l\}, \ldots, \{p(\bar{c}, t) \mid l \cdot \lfloor \frac{c_{\max}}{l} \rfloor \le t \le c_{\max}\}$. For each slice, we independently apply our MT-Diff-Learn framework to learn locally valid rules. Finally, all extracted rules across slices are aggregated into a single file. Note that using this algorithm, we identify besides the window size also the slice size as a hyperparameter.

# 4   EXPERIMENTAL RESULTS

We showcase the applicability of our approach to three different scenarios. We first consider link prediction in Temporal Knowledge Graphs (tKG), then we show how we can exploit the expressibility of our output language to capture the common request-grant schema. In our experiments, we are interested to see (Q1) whether the increased expressiveness can be fruitfully leveraged, (Q2) scalability and succinctness of the approach, and (Q3) how answer performance is affected.

**Experimental Setup**   We have implemented our approach in an experimental prototype, MT-Diff-Learn, that is available in the supplementary material.[1]  For the experiments, we used a platform with 224GB of memory. All experimental set-ups follow the same schema, viz. the standard evaluation protocol with temporal filtering Han et al. (2020). Each temporal dataset is split into training, validation, and test subsets according to the temporal constraints imposed. From the training and validation data, we extract a DatalogMTL program $\pi$ and a scoring function $\tau : \pi \to [0, 1]$ assigning a weight to each rule in $\pi$, which is learned during training as discussed above; it reflects the relevance of rule $r$ for capturing temporal patterns in the data.

As for (Q1), we inspect whether (where applicable) meaningful expected rules are generated. For (Q2), we use the number of rules generated to measure model size, and the resources needed. Regarding (Q3), we use two standard metrics *mean reciprocal rank (MRR)* and *Hit@k*, for $k \in \{1, 10\}$ as in (Wang et al. 2024). [2]  Both MRR and Hit@k refer to $rank_i$, which is the position of the correct answer in the ordered list of answers to a query $q_i$; if the correct answer does not appear, $rank_i = \infty$.

More in detail, for each test fact $R(a, b)@t$, we construct the queries $R(x, b)@t$ and $R(a, x)@t$, where $a$ resp. $b$ is an expected answer, and let the dataset $D$ include all training and validation facts. We then compute all constants $c$ such that $\pi[D]$ contains $r(c, b)$ at $t$, denoted $R(c, b)@t \in T_\pi(D)$, as candidate answers. The score of $c$ is defined as $\max\{\tau(r) \mid r \in \Pi, R(c, b)@t \in r[D]\}$ i.e., the maximum score among the rules deriving $R(c, b)@t$.

Candidates are sorted descendingly by score, breaking ties first by considering secondary rule scores and then alphabetically. We apply *temporal filtering* Han et al. (2020); Liu et al. (2022) to remove each candidate $c \ne a$ such that $R(c, b)@t$ appears in the training, validation, or test set. The rank of the correct answer is its position in the filtered list. For $R(a, x)@t$ queries, we proceed analogously.

 **Link Prediction on Temporal Knowledge Graphs**   In missing link prediction, we aim to predict which facts belong to the completion $D^*$ of a given tKG as dataset $D$. Specifically, we address queries generated from $R(a, b)@t$ as above, where $t_{\max}$ is the maximum timestamp in $D$; i.e., find substitutions for $x$ that make the query true in $D^*$ in the observed interval (known as interpolation).

---

[1]Currently for arities less or equal 2; an extension to all arities is simple wich unchanged neural design.

[2]We notice some inconsistent inference issues in MTLearn's original code; see Appendix for details.

| Model | ICEWS14 MRR | H@1 | H@10 | ICEWS0515 MRR | H@1 | H@10 |
|---|---|---|---|---|---|---|
| TTransE (Bordes et al., 2013) | 25.5 | 7.4 | 60.1 | 27.1 | 8.4 | 61.6 |
| TADistMult (Lin et al., 2023) | 47.7 | 36.3 | 68.6 | 47.4 | 34.6 | 72.8 |
| LCGE$^+$ | 61.6 | 53.2 | 77.5 | 61.8 | 51.4 | 81.2 |
| TLT-KGE + HGE$^+$ | 63.0 | 54.9 | 77.7 | 68.6 | 60.7 | 83.1 |
| 50k-MTLearn | 8.9 | 5.5 | 16.5 | – | – | – |
| 200k-MTLearn | 12.4 | 7.6 | 22.9 | – | – | – |
| 500k-MTLearn | 17.6 | 12.8 | 28.3 | – | – | – |
| **MT-Diff-Learn** | _38.3_ | _32.7_ | _45.2_ | 42.5 | 37.8 | 50.0 |

Table 1: Interpolation performance on ICEWS14 and ICEWS0515. Results with $^+$ come from (Pan et al., 2024). The underline / dash marks the best rule-based result / second-best result. MT-Diff-Learn was run with: breadth $b = 8$, depth $d = 1$, window_size $w = 5$, slice=44 and $\beta = .5$

| Model | Test Set (45 data) MRR | H@1 | H@10 | #Rules |
|---|---|---|---|---|
| **MT-Diff-Learn** | 54.46 | 27.03 | 100.0 | 168 |
| MTLearn | 49.2 | 26.5 | 100.0 | 111,649 |

Table 2: Performance on the action cyber-physical scenario

| Model | Test Set (45 data) MRR | H@1 | H@10 | #Rules |
|---|---|---|---|---|
| **MT-Diff-Learn** | 42.2 | 20.0 | 88.8 | 183 |
| MTLearn | 40.6 | 19.1 | 100.0 | 31,391 |

Table 3: Performance on the action description scenario

State-of-the-art approaches to temporal link prediction are embedding-based Leblay & Chekol (2018); Lacroix et al. (2020); Xu et al. (2021), encoding entities, relations, and timestamps into a vector space and reasoning via latent representations. They achieve strong predictive performance but are inherently opaque and thus troubled in domains where explainability is essential. Table 1 compares MT-Diff-Learn with MTLearn and embedding-based systems on the interpolation task. The public MTLearn pipeline produces about 21M rules, while our model yields about $40,000$. To enable evaluation on our hardware for ICEWS14, we assess sampled variants of MTLearn ($n$k-MTLearn). For ICEWS05-15, rule generation with the released MTLearn pipeline did not complete under the evaluation setup in Wang et al. (2024). These observations suggest that reproducing the reported MTLearn scores may require exceptionally large rule sets, with implications for scalability.

**Cyber-Physical Scenario**  To test our approach to cyber-physical settings, we created a synthetic dataset simulating usual access-request and access-grant interactions among entities (institutions), where $request_k(i,j)$ and $grant_k(i,j)$, $k = 0, 1$ mean that institution $i$ request access to resource $k$ from institution $j$ resp. $i$ grants access to $k$ to $j$. The generation is challenging due to the noise injected into the data. E.g., not every request may be followed by a grant, and some grants may have no preceding request. However, the noise is small enough to warrant that most generated data preserves the causal relation. If a request like $request_0(81, 14)@1$ was generated, then $grant_0(14, 81)@(1+d)$ is generated with a delay $d$ sampled from a normal distribution with mean 7 and standard deviation 2, capturing realistic temporal dynamics.

The queries are generated from $grant_k(i, j)@t$ as above. The results in Table 2 show that MT-Diff-Learn, which leverages the $\lozenge$ and $\blacklozenge$ (Diamond) operators to capture the occurrence of requests within an interval, significantly outperforms MTLearn. Using only the $\square$ and $\blacksquare$ (Box) operators leads to overfitting on the training data if compared to the results from the test cases.

**Action Description Scenario**  We enhance the previous setting to illustrate how temporal rules may aid finding action descriptions, which is an important modeling task. Ideally, we aim to extract two types of rules from the neural model: (1) those describing the effects of actions and (2) those capturing conditions that trigger action execution, which is a strong version of preconditions.

For each fact $grant_k(i,j)@t$ in the previous dataset, we added a fact $access_k(j,i)@(t+1)$ stating that $j$ has access to resource $k$ of $i$ in the next timepoint. The queries are obtained from such facts. The performance is shown in Table 3. As regards (Q1), in our setting, we are able to learn rules like

$$access_k(X, Y) \leftarrow \blacksquare_{[1,1]} grant_k(Y, X).$$

modeling a direct effect of granting access. Further, we can learn rules

$$grant_k(Y, X) : -\blacklozenge_{[4,10]} request_k(X, Y)$$

that granting happens with delay of 4 to 10 time units after request, modeling temporal uncertainty. The higher performances in Table 3 compared to Table 2 from MTLearn showcase its ability to succesfully capture local deterministic temporal patterns such the effect of an action. Note that circa 45 thousand rules were produced, while using MT-Diff-Learn, only around 80. Ablation results in the Appendix demonstrate that removing the eventuality operator causes a sharp performance drop.

## 5 RELATED WORK AND CONCLUSION

Closest to our work is MT-Learn (Wang et al. 2024), which employs AnyBurl (Meilicke et al., 2019), a bottom-up atemporal rule learner. AnyBurl samples ground paths of length $k$ in a knowledge graph and generalizes them by replacing entities with variables. Sampling continues until newly generated rules are no longer novel (a novelty ratio threshold is used). This check is purely syntactic, and imposing ordering on variables helps in containing the number of first-order rules it produces. Once the novelty threshold is met, the path length is increased to $k + 1$, and the process repeats. Such bottom-up approaches tend to generate a large number of rules, as observed in the application of MTLean on realistic datasets. In contrast, our framework learns directly rules at the first-order level. We first lift the ground atoms to first-order atoms, and then search for a limited set of rules that aim to cover the ground atoms they represent.

Related to our work is LTL specification learning by Ielo et al. (2023), who, however, did not consider metric atoms and used ILASP, a SOTA tool for symbolic rule learning. Most approaches to link prediction rely on neural architectures or embedding techniques augmented with temporal dimensions, among them RE-Net Jin et al. (2019), TTransE Leblay & Chekol (2018), TA-DisMult (Garcia 2018), TeLM Xu et al. (2021), TComplEx (Lacroix et al. 2020), and LCGE Niu & Li (2023). Our approach instead extracts interpretable metric rules that directly model temporal dependencies without relying on latent representations. The TLogic framework (Liu et al. 2022) produces rules that in contrast to ours can not handle interpolation and are thus limited in expressiveness.

Our ongoing work aims to support dense timelines, higher predicate arities, and to extend the language with flat function terms to increase readability and performance.

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

# A  APPENDIX

In what follows, the experiments

- were conducted on a high-performance server equipped with $2\times$ Intel Xeon Gold 5416S CPUs (16 cores, 2 threads per core, $2.00\,\mathrm{GHz}$), $8\times$ NVIDIA L40S GPUs (48 GB VRAM each), and 2 TB of system memory;

All experimental data and code will be made available upon possible acceptance. The reviewers may access the source code at the link provided in the last page under the "SOURCE CODE AND DATA" section of the Appendix.

## A.1  RESULTS ON THE INTERPOLATION-LINK PREDICTION ON TEMPORAL KNOWLEDGE GRAPHS

The statistics about the datasets that we considered are shown in Table 1.

| Dataset | #Relations | #Entities | #Facts |
|---|---|---|---|
| ICEWS14 | 231 | 7,130 | 72,826 |
| ICEWS05-15 | 252 | 10,489 | 368,962 |

Table 1: Statistics of the temporal knowledge graph datasets.

**MTLearn**   We attempted to replicate the results reported for MTLearn (Wang et al. 2024). Specifically, we successfully generated rules for:

1. the interpolation task of ICEWS14.

However, for ICEWS05-15, the provided code encountered an array index bug, preventing successful rule generation. We contacted the developer regarding this issue, but he indicated that he is no longer working in academia and therefore could not provide detailed support.

The number of rules produced for the successful runs was:

- ICEWS14: $21,152,956$ rules

while MT-Diff-Learn produced 41,677 rules in the best run. Note that this number is more than two order of magnitudes smaller than the number of rules produced by MTLearn (in fact, about 0.2% of the rules produced MTLearn).

The original evaluation script also contained several bugs, some of which were fixable, while others were more involved due to dependencies on the `https://pypi.org/project/ meteor-reasoner/` Python library. Moreover, due to the exceptionally large number of rules, our evaluation script ran out of memory, preventing the computation of evaluation metrics for these tasks on our machine. The comparison therefore is tricky, as our tool produces a number of rules at most linear in the number of data from the training set. Therefore, in the next section, we introduce a sampled version of the MTLearn output.

**Resource-Bound MTLearn**   In Table 4, we show the best results, the mean, and the standard deviation of three different settings in which we were able to run on our machine. We produced the results of MTLearn for window size 8 and scoring strategy maximum, the default options of the tool; we sampled respectively 50K, 200K, and 500K rules. We executed this procedure for three times, and we computed the metrics.

**MT-Diff-Learn**   Recall that $n_r^h$ from the **Neural Network** paragraph stands for the number of rules used to derive the target first-order atom $h$. TWe always get a number of rules at most equal to the number of facts that are reported in Tables 2 and 3. In the experiments discussed in the following sections, we also report the number of rules so that, even if the framework produces a linear number of rules by design, we also show some empirical evidence of this fact.

| Metric | MT-Diff-Learn | | |
|---|---|---|---|
| | MRR | Hit@1 | Hit@10 |
| Best | 38.3 | 32.7 | 45.2 |
| Mean | 34.1 | 29.5 | 43.2 |
| Std. Deviation | 2.5 | 2.2 | 0.9 |
| Num_Rules (Best) | 39, 197 | | |
| Num_Rules (Mean) | 41,758.7 | | |
| Num_Rules (Std. Deviation) | 82.0 | | |

Table 2: Performance summary and number of rules for MT-Diff-Learn on ICEWS14.

| Metric | MT-Diff-Learn | | |
|---|---|---|---|
| | MRR | Hit@1 | Hit@10 |
| Best | 42.5 | 37.8 | 50.0 |
| Mean | 42.0 | 37.7 | 49.8 |
| Std. Deviation | 0.7 | 0.1 | 0.6 |
| Num_Rules (Best) | 210,880 | | |
| Num_Rules (Mean) | 210,997.3 | | |
| Num_Rules (Std. Deviation) | 111.5 | | |

Table 3: Performance summary and number of rules for MT-Diff-Learn on ICESW05-15.

| Metric | 500k-MTlearn | | | 200k-MTlearn | | | 50k-MTlearn | | |
|---|---|---|---|---|---|---|---|---|---|
| | MRR | Hit@1 | Hit@10 | MRR | Hit@1 | Hit@10 | MRR | Hit@1 | Hit@10 |
| Best | 17.6 | 12.8 | 28.3 | 12.4 | 7.6 | 22.9 | 8.9 | 5.5 | 16.5 |
| Mean | 15.64 | 10.62 | 26.63 | 11.86 | 7.06 | 22.39 | 8.41 | 5.17 | 15.82 |
| Std. Deviation | 2.28 | 2.24 | 2.41 | 0.68 | 0.51 | 0.64 | 0.47 | 0.35 | 0.55 |

Table 4: Performance statistics of MTLearn in link prediction for different dataset sizes

## A.2   RESULTS ON THE CYBER-PHYSICAL SETTING

| Metric | MT-Diff-Learn | | | | MTLearn | | | |
|---|---|---|---|---|---|---|---|---|
| | MRR | Hit@10 | Hit@1 | #Rules | MRR | Hit@10 | Hit@1 | #Rules |
| Best Result | 56.0 | 95.2 | 34.5 | 84 | 43.3 | 100.0 | 20.8 | 9,564 |
| Mean Result | 53.8 | 92.9 | 31.7 | 85.3 | 41.5 | 100.0 | 20.5 | 12,295 |
| Std. Deviation | 3.8 | 4.1 | 3.8 | 2.3 | 100.0 | . | 0.6 | 2,371.1 |
| Dataset Gen. $(\mu, \sigma)$ | Mean = 7, Std Deviation = 2, Size Training Set = 202 | | | | | | | |

Table 5: Comparison of MT-Diff-Learn and MTLearn in the cyber-physical scenario, dataset generation with $\mu = 7$ and $\sigma = 2$. Window size: 10 for both.

| Metric | MT-Diff-Learn | | | | MTLearn | | | |
|---|---|---|---|---|---|---|---|---|
| | MRR | Hit@10 | Hit@1 | #Rules | MRR | Hit@10 | Hit@1 | #Rules |
| Best Result | 54.5 | 100.0 | 32.4 | 165 | 46.7 | 1.0 | 22.4 | 11,8041 |
| Mean Result | 47.6 | 91.9 | 23.4 | 174 | 46.0 | 1.0 | 22.0 | 11,8497.3 |
| Std. Deviation | 6.4 | 7.2 | 8.8 | 8.5 | 1.2 | . | 0.6 | 455.5 |
| Dataset Gen. $(\mu, \sigma)$ | Mean = 15, Std Deviation = 2, Size Training Set = 196 | | | | | | | |

Table 6: Comparison of MT-Diff-Learn and MTLearn in the cyber-physical scenario, dataset generation with $\mu = 15$ and $\sigma = 2$. Window size: 35 for both.

| Metric | MT-Diff-Learn | | | | MTLearn | | | |
|---|---|---|---|---|---|---|---|---|
| | MRR | Hit@10 | Hit@1 | #Rules | MRR | Hit@10 | Hit@1 | #Rules |
| Best Result | 54.5 | 100.0 | 27.0 | 168 | 49.2 | 100.0 | 26.5 | 111,649 |
| Mean Result | 51.6 | 99.1 | 26.1 | 174 | 48.7 | 100.0 | 25.7 | 111,650 |
| Std. Deviation | 0.80 | 1.5 | 1.5 | 6 | 0.7 | 0.0 | 1.0 | 1.41 |
| Dataset Gen. $(\mu, \sigma)$ | Mean = 15, Std Deviation = 5, Size Training Set = 189 | | | | | | | |

Table 7: Comparison of MT-Diff-Learn and MTLearn in the cyber-physical scenario, dataset generation with $\mu = 15$ and $\sigma = 5$. Window size: 35 for both.

**MTLearn**   MTLearn was run using the same window-size parameters as MT-Diff-Learn, namely: window size of 10 when the mean is 7, while window size of 35 when the mean is 15. and with a scoring strategy of maximum, which is the default option.

**MT-Diff-Learn**   The model was trained using the following parameters: number of epochs 20, breadth 6, and depth 1.

We further show results that we obtained for variations of the dataset generation with different distribution and window size in Tables 5–7. As one can see, then number of rules produced by MT-Diff-Learn grows linearly in the size of the dataset, while it is still able to outperform MTLearn in different scenarios. This analysis showcases the ability of MT-Diff-Learn for scalability.

## A.3 RESULTS ON THE ACTION DESCRIPTION TASK

| Metric | MT-Diff-Learn | | | | MTLearn | | | |
|---|---|---|---|---|---|---|---|---|
| | MRR | Hit@10 | Hit@1 | #Rules | MRR | Hit@10 | Hit@1 | #Rules |
| Best Result | 34.1 | 56.3 | 22.4 | 128 | 42.08 | 94.83 | 19.8 | 23,964 |
| Mean Result | 32.1 | 53.5 | 19.4 | 179.3 | 41.52 | 92.7 | 20.5 | 23,964 |
| Std. Deviation | 2.48 | 2.40 | 2.86 | 45.17 | 0.97 | 1.82 | 0.57 | 10.0 |
| Dataset Gen. $(\mu, \sigma)$ | Mean = 7, Std Deviation = 2, Size Training Set = 299 | | | | | | | |

Table 8: Comparison of MT-Diff-Learn and MTLearn in the action-description scenario, dataset generation with $\mu = 7$ and $\sigma = 2$. Window size: 10 for both.

| Metric | MT-Diff-Learn | | | | MTLearn | | | |
|---|---|---|---|---|---|---|---|---|
| | MRR | Hit@10 | Hit@1 | #Rules | MRR | Hit@10 | Hit@1 | #Rules |
| Best Result | 39.6 | 87.5 | 19.4 | 190 | 46.7 | 99.6 | 27.6 | 23,977 |
| Mean Result | 38.7 | 83.3 | 18.5 | 183.3 | 49.4 | 98.6 | 24.5 | 24,048.3 |
| Std. Deviation | 1.5 | 3.8 | 0.1 | 7.02 | 2.8 | 1.0 | 2.7 | 117.5 |
| Dataset Gen. $(\mu, \sigma)$ | Mean = 15, Std Deviation = 2, Size Training Set = 283 | | | | | | | |

Table 9: Comparison of MT-Diff-Learn and MTLearn in the action-description scenario, dataset generation with $\mu = 15$ and $\sigma = 2$. Window size: 35 for both.

| Metric | MT-Diff-Learn | | | | MTLearn | | | |
|---|---|---|---|---|---|---|---|---|
| | MRR | Hit@10 | Hit@1 | #Rules | MRR | Hit@10 | Hit@1 | #Rules |
| Best Result | 42.2 | 88.8 | 20.0 | 183 | 40.6 | 100.0 | 19.1 | 31391 |
| Mean Result | 42.0 | 85.4 | 17.5 | 180.7 | 39.7 | 100.0 | 16.4 | 31655 |
| Std. Deviation | 2.6 | 3.1 | 2.2 | 2.1 | 1.0 | 0.0 | 2.3 | 228.6 |
| Dataset Gen. $(\mu, \sigma)$ | Mean = 15, Std Deviation = 5, Size Training Set = 285 | | | | | | | |

Table 10: Comparison of MT-Diff-Learn and MTLearn in the action-description scenario, dataset generation with $\mu = 15$ and $\sigma = 5$. Window size: 35 for both.

**MTLearn** MTLearn was run using the same window-size parameters as MT-Diff-Learn, namely: window size of 10 when the mean is 7, while window size of 35 when the mean is 15. and with a scoring strategy of maximum, which is the default option.

**MT-Diff-Learn** The model was trained with the following parameters: number of epochs 20, breadth 6, and depth 1. The window size parameter coincides with that of the MTLearn runs.

We further show results that we obtained for variations of the dataset generation with different distribution and window size in Tables 8–10. As one can see, also in this scenario, our model is able to produce acceptable performances keeping the number or rules limited.

### A.4 ABLATION STUDY ON THE CYBER-PHYSICAL SETTING

We investigate the contribution of the eventuality operators by conducting an ablation study in the cyber-physical scenario. Specifically, we compare the performance of MT-Diff-Learn when both eventuality operators ($\Diamond$, $\blacklozenge$) are available against a variant that replaces them with the always operator ($\Box$, $\blacksquare$). Note that in both settings, punctual intervals are still permitted. This highlights the importance of temporal evolutions for capturing system dynamics, where the grant occurs with uncertainty: it is expected to happen within an interval, but the exact time point remains unknown. Note also that the request-implies-grant is a canonical pattern in such scenarios.

| Metric | MT-Diff-Learn-with-$\{\Diamond, \blacklozenge\}$ | | | MT-Diff-Learn-with-$\{\Box, \blacksquare\}$ | | |
|---|---|---|---|---|---|---|
| | MRR | Hit@10 | Hit@1 | MRR | Hit@10 | Hit@1 |
| Best Result | 54.46 | 100.0 | 32.43 | 15.5 | 18.9 | 12.2 |
| Mean Result | 47.60 | 91.89 | 23.42 | 13.7 | 15.3 | 12.2 |
| Std. Deviation | 6.43 | 7.15 | 8.79 | 1.6 | 3.1 | 0 |
| Dataset Gen. $(\mu, \sigma)$ | Mean = 15, Std Deviation = 2, Size Training Set = 196 | | | | | |

Table 11: Comparison of MT-Diff-Learn and MTLearn in the cyber-physical scenario, dataset generation with $\mu = 15$ and $\sigma = 2$. Window size: 35 for both.

## B SOURCE CODE AND DATA

The source code and data can be found here:
https://drive.google.com/drive/folders/12sJDhN1nGZ49iGJEFKmm8VGYsr5laSq7?usp=sharing.

## C REBUTTAL

### C.1 FIRST REVIEWER

**Weaknesses** >> The paper is a bit like reading someone's half-documented code. The paper says "We do this, then we do that, then we do that." But it is often not clear what the goal is or why those steps are appropriate. This is particularly a problem in section 3. It also makes it difficult to adjust for lapses in the exposition. To take some early examples: "target atom" isn't defined at line 126 and it's not clear where it comes from or why it's needed.

Thanks for your comments. We noticed this issue and updated the presentation in a more schematic style. Please check Section 3 in the revised version.

Regarding the conceptual clarifications: our objective is to learn a logic program $P$ that derives a designated *target atom* $h$, given background knowledge $B$ and sets $\mathcal{P}$ and $\mathcal{N}$ of positive and negative examples. A valid solution $P$ must entail all positive examples while excluding all negative ones:

$$B \cup P \models e^+ \ \forall e^+ \in \mathcal{P}, \qquad B \cup P \not\models e^- \ \forall e^- \in \mathcal{N}.$$

Instead of relying on a black-box neural predictor, the network is structured so as to learn a program $P$ whose inference behaviour follows the formal semantics of DatalogMTL. Predictions are framed as logical entailment: the model determines whether the target atom (i.e., the output label $y$) holds.

For example, if *wet_ground* is the target atom, the system may learn a rule such as:

$$wet\_ground \ \leftarrow \ \blacklozenge[2, 0] \ rains,$$

which, under the semantics of DatalogMTL, states that the ground is wet if it has rained at any point within the last two days or on the current day in a scenario where the time unit is a day. In general, *target atoms* are precisely those atoms whose truth values the model aims to predict via the learned logical rules.

>> At line 132, it's not clear where the threshold comes from: before relaxation, is it meant to be the number of negative literals in the rule?

We wrote: "Let $v$ be an real number, and let $\varphi(v) = v'$ be the threshold function, where $v' = 1$ if $v \geq \tau$, and $v'_i = 0$ otherwise with $\tau$ being the threshold value." Instead of $v'_i = 0$, it should have

been $v' = 0$. Perhaps the typo hindered the reading. Therefore, it is not the number of negative literals. It is a threshold value such that if a value is greater than $\tau$, then it collapses to 1, otherwise to 0.

$>>$ At line 202, what happens if multiple atoms were "already considered in the previous windows"?

If multiple atoms were already considered in previous windows, one of them is selected uniformly at random. The remaining atoms can then still be used in the current window to *represent* distinct ground instances. For example, suppose both $consult(X, Y)$ and $consult(W, Z)$ were previously generated. Then, if the current window contains the ground atoms $consult(a, b)$ and $consult(c, d)$, we may map $consult(X, Y)$ to represent $consult(a, b)$ while $consult(W, Z)$ remains available to represent $consult(c, d)$.

$>>$ Even as a low-level description of what's done, the paper doesn't quite hang together. To take one example: Line 190 defines the "overlap score" and says that it is "used to determine processing order," but it's not clear what "processing order" refers to, and neither nor the overlap score is ever mentioned again! (I suppose it's the same as the "co-occurring score" in the next paragraph.)

Thank you for pointing this out. Our heuristics compute two different scores: (i) an overlap score used to determine the order in which windows are processed, and (ii) a co-occurrence score used to decide the order in which ground atoms are lifted to first-order ones. We intended the textual description to be easier to follow than presenting a full procedural algorithm, but we see how this may have caused confusion. If allowed, we will include a clear algorithmic description in the appendix to make the distinction explicit.

$>>$ Furthermore, the definition of contains unbound variables. (I suppose it should have been named .)

Thank you for highlighting this point. As stated in the paper, the number of rules we generate is linear in the input, which implies that the number of variables is indeed bounded. We made it clearer by exposing the procedure in an algorithmic fashion: the generation of the variables is described from line 14 to line 17 in Algorithm 1. One cannot generate more $|E||\mathcal{W}|$ variables, where $E$ denotes the set of all entities and $|\mathcal{W}|$ the number of windows.

**Questions**   $>>$ What is the formal learning problem here?

The formal learning problem we address is that of learning DatalogMTL rules (defined in Section 2) that describe a temporal dataset. More precisely, our setting corresponds to the temporal interpolation problem studied in prior work such as [1] and [2]: given temporal sequences of facts, the goal is to induce a rule set whose temporal consequences reconstruct or generalize these sequences.

[1] Zhen Han, Peng Chen, Yunpu Ma, and Volker Tresp. Explainable subgraph reasoning for forecasting on temporal knowledge graphs. In the International conference on learning representations, 2020.

[2] Yushan Liu, Yunpu Ma, Marcel Hildebrandt, Mitchell Joblin, and Volker Tresp. Tlogic: Temporal logical rules for explainable link forecasting on temporal knowledge graphs. In Proceedings of the 36th AAAI Conference on Artificial Intelligence (AAAI 2022), pp. 4120–4127. AAAI Press, 2022.

$>>$ Is it a machine learning problem where there is a true set of temporal facts over intervals, the answers to some queries are observed at random, and the estimated DatalogMTL program should accurately predict the answers to other queries?

The learning problems we consider are indeed machine-learning tasks over temporal data, but they span three concrete settings rather than a single abstract formulation:

1. Temporal link prediction (interpolation) on temporal knowledge graphs, following the setting of [1] and [2], where the goal is to reconstruct unseen temporal facts from partially observed sequences.

2. A synthetic temporal dataset in which the model must learn temporal relations connecting requests and grants.

3. A temporal action-learning scenario, where the system induces action descriptions from observations of an autonomous agent.

More generally, we learn a DatalogMTL program that generalizes from observed temporal facts to unobserved ones. We tested it across different temporal reasoning domains to show its versatility.

$>>$ Is it a statistics problem where a true parameter is a DatalogMTL program, the answers to some queries are observed at random, and the true program should be recovered in the limit? If so, is the program in fact identifiable?

In the ICEWS settings, there is no clear, perfect program one should learn in the limit, while in the synthetic dataset, yes. For instance, the rules presented in the Action Description Scenario setting represent some possible *good* rules one would like to learn given the parameters used in the generation of the data.

$>>$ Is it a computational problem of finding a small description of a set of positive facts

It does not necessarily need to be small, however, if it is small, it is preferrable. Our tool learns indeed a small number of rules (linear) in the size of the dataset.

$>>$ (and if so, how is it ensured that the description does not predict false facts, since negative examples are not provided)?

In the [1] and [2] settings, an answer is always expected.

[1] Zhen Han, Peng Chen, Yunpu Ma, and Volker Tresp. Explainable subgraph reasoning for forecasting on temporal knowledge graphs. In the International conference on learning representations, 2020.

[2] Yushan Liu, Yunpu Ma, Marcel Hildebrandt, Mitchell Joblin, and Volker Tresp. Tlogic: Temporal logical rules for explainable link forecasting on temporal knowledge graphs. In Proceedings of the 36th AAAI Conference on Artificial Intelligence (AAAI 2022), pp. 4120–4127. AAAI Press, 2022.

$>>$ Is it an association rule mining problem as mentioned earlier? If so, what is the success criterion?

The success criterion is not based on association-rule mining but on standard evaluation metrics used in temporal link prediction and temporal reasoning tasks. In all experiments, we assess performance using Hits@Rank and MRR, which measure how well the learned DatalogMTL program predicts the correct temporal facts among all candidate answers.

$>>$ What does "consecutive occurrences" mean at line 181? You talk about t, t+1, and t+2. I had assumed that time was continuous, since you referred early on to "intervals"; did you actually intend for time to be discrete?

Yes, in our experimental setting time is discrete. This allows us to refer to consecutive occurrences such as $t$, $t + 1$, and $t + 2$, which correspond to the discrete interval $[t, t + 2]$. Intervals can still be defined naturally over a discrete timeline. Extending the framework to continuous time is an interesting direction for future work; it would mainly require adjustments in data preprocessing rather than changes to the core method.

$>>$ It's not clear to me whether learning temporal logic programs is an important problem. Can you make a case for it?

Learning temporal logic programs is important because many real-world reasoning tasks are inherently temporal: events have durations, actions have delayed effects, and relations evolve over time. The ability to induce temporal rules from data places our work within a growing line of research in Inductive Logic Programming (ILP) and neuro-symbolic reasoning, where temporal extensions are becoming increasingly relevant.

Temporal logic programs provide high-level, human-readable rules that capture how facts evolve over time. These rules are valuable not only for prediction (e.g., temporal link prediction, learning action descriptions) but also for explainability and diagnosis in evolving systems, where extracting structured knowledge from data is crucial. In contrast, transformer-based models—despite recent efforts to improve their interpretability, remain inherently more opaque than symbolic formalisms,

which offer explicit reasoning steps. Symbolic approaches may sometimes trade off raw performance, but they provide semantic clarity that is essential in domains where understanding temporal dynamics matters.

$\gg$ But as an alternative, perhaps one could train a neural generative model and then extract explanatory patterns from it post hoc.)

Training a neural generative model and extracting patterns post hoc is indeed a possible alternative. However, our goal is to extract rules that correspond directly to the actual decision-making process of the model—i.e., explicit if–then clauses that the system truly relies on during inference. Post-hoc pattern results in a knowledge that does not offer the same properties of a formal language with a clear syntax and semantics. Therefore it cannot directly support classical knowledge-representation tasks such as logical inference, automated planning, verification, or diagnosis. In contrast, our method learns rules within a formal logic framework (DatalogMTL), ensuring that the extracted rules can be given as an input to temporal rule solvers or monitoring frameworks.

To integrate: "post-hoc interpretation, we take an existing machine learning system, that has already been trained, and try to understand its inner state. In the other approach, designing explicit already-interpretable machine learning systems, we con- strain the design of the machine learning system to guarantee, in advance, that its results will be interpretable"

$\gg$ I am not able to follow the details of the method as presented. On first principles, I would have expected a method similar to Gao et al. (2024), which you present as your starting point, but where the matrix columns had names like (for all T in [Time+a,Time+b]) property(T,X,Y) or (exists T in [Time+a,Time+b]) property(T,X,Y). These truth conditions of such a column would be softened, in part by fuzzing the edges of the interval [Time+a,Time+b]. Thus, you could improve a, b by following their gradient. Why didn't you do it this way?

We may be misunderstanding the reviewer's suggestion, but our architecture does already allow for differentiable learning of interval endpoints. In DatalogMTL, temporal operators such as boxes or diamonds are implemented through differentiable layers, and the parameters are learned via gradient descent— analogous in the same spirit the reviewer suggests.

$\gg$ What happens if you make the window size too large or too small?

The size of the windows is optimized via a grid-search. The idea is that the windows should be large enough to capture most of the relevant temporal dependencies across the data.

$\gg$ Does this change the number of rules you find and their specificity, so that you might underpredict or overpredict positive facts?

Changing the window size can indeed influence the specificity of the learned rules. Intuitively, larger windows rely less on temporally local dependencies and may therefore lead to more general rules, which can increase the risk of overgeneralization. On this other side, when the temporal dependencies in the data are short-term, increasing the window size may introduce unnecessary complexity and this may negatively affect the quality of the learned rules. We will include this discussion in the revised version to clarify how window size interacts with rule specificity and prediction accuracy with some ablation studies to support this.

$\gg$ What are the simplest examples where your heuristics would fail?

The simplest example is when every entity appears only once, and the first-order "representatives" do not generalize. In fact, in this case, the outcome will be a grounded DatalogMTL program.

## C.2 SECOND REVIEWER

**Weaknesses** $\gg$ The experimental section could be strengthened by including comparisons against more diverse state-of-the-art neuro-symbolic or temporal rule mining baseline approaches, beyond standard non-temporal or limited temporal inductive logic programming methods, to fully contextualize the proposed model's performance in the broader field of temporal sequence modeling.

We interpret the reviewer's comment as referring to temporal sequence modeling. Our focus, however, is on interpretable rule learning, not black-box sequence modeling. While sequence models such as LSTMs or Transformers can handle temporal prediction, they do not produce symbolic rules

or interpretable structures, making them unsuitable baselines for our setting. Of course, the literature is very huge, so we reported only some results of temporal sequence modeling tools in the Table about temporal link prediction in the interpolation settings. Nonetheless, we will clarify this distinction in the paper.

**Questions** $>>$ How robust is the learning of the continuous metric interval bounds to noise in the training data,

Our syntactic dataset is obtained by adding some uncertainty on the realization of the grant via the parameters $\mu$ and $\sigma$, and some noise is injected as the grant is given if a uniform probability variable that uniformly ranges from 0 to 1 exceeds .9.

$>>$ and what regularization or loss terms (if any) are specifically implemented to prevent interval collapse or explosion during gradient descent?

We have not considered such regularization.

$>>$ Given the focus on interpretability, can the authors provide a more detailed analysis of the learned rules—perhaps a qualitative summary or examples from the different use cases—to illustrate how MT-Diff-Learn discovers non-obvious or complex temporal relationships that purely sequential models might miss?

An example of a rule that is learned in the ICEWS setting is:

$$consult(X, Y) \leftarrow \square_{[0,0]} consult(Y, X).$$

This rule is not di per see involving a temporal complex structure, but it showcase how certain property such as symmetry in this case can be easily represented.

A temporal which spans over different timestamps is the following:

$$sign\_agreement(X, Y) \leftarrow \blacksquare_{[1,2]} consult(Y, X).$$

This rule is rather general, as it can be instantiated in many different ways. Its intended meaning is the following: at a time point $t$, the atom $sign\_agreement(X, Y)$ holds whenever subject $b$ has been consulting subject $a$ *throughout* the interval $[t-2, t-1]$. In other words, if $b$ has been consulting $a$ during the previous day and the day before, then at time $t$ subject $a$ will sign an agreement with $b$.

### C.3 THIRD REVIEWER

**Weaknesses** $>>$ The lifting process is complex and relies heavily on heuristics (overlap scores, processing order, variable assignment constraints). The robustness of the system to these choices is unclear, and the impact of associated hyperparameters (window size $l$, breadth $b$, depth $d$) is not analyzed. The generalizability of this lifting process warrants further investigation.

We agree that more ablation studies should be provided, and we aim to integrate them in the revised version of the paper.

$>>$ On standard tKG benchmarks (ICEWS), MT-Diff-Learn significantly lags behind SOTA embedding-based methods. While the focus is on interpretability, this gap may limit adoption where accuracy is paramount. The paper should better discuss this trade-off.

We agree that embedding-based temporal KGC methods achieve higher raw accuracy on large-scale benchmarks such as ICEWS. However, these models are inherently black-box and do not yield interpretable temporal rules. We will clarify this trade-off in the revised version and position our approach as an interpretable alternative rather than a direct competitor to high-performance embedding models. Furthermore, in cases where the labels of entities and relation names do not carry any semantic information, the performance of embedding-based approaches may degrade significantly, whereas rule-based methods remain robust to such semantic neutrality.

$>>$ Key parts of the methodology are difficult to parse. The lifting procedure (W1) is confusing. Furthermore, the process of translating the learned intervals back into DatalogMTL syntax is unclear. The example provided (transforming [2, 5] into a conjunction of past and future operators) seems overly complicated and requires clarification regarding how semantics are preserved relative to the window center.

The translation from learned intervals into DatalogMTL is conceptually simple, so that part should be made more readable if it causes any problem.

$>>$ The abstract claims applicability to data over dense time intervals. However, the semantics defined in Section 2 are explicitly over integers ($\mathbb{Z}$), and dense time is only mentioned as future work.

Yes, with some adaptation in the preprocessing of the data we can apply this method to learn also rules operating on a dense timeline, however, to ease the presentation we stick to the simpler case of the integers. We will make this clearer in the revised version.

$>>$ While the output model is succinct, the training process involves complex tensor operations. The input tensors (Start/End) have a dimension representing the maximum number of disjoint intervals an atom might hold in a window. The paper does not detail the practical implications of this on memory usage and training time if this value is large.

For each target atom $h$, we generate the three tensors for (i) the truth matrix, (ii) the starting and (iii) the ending matrix. Since the number of the relevant atoms is parametrized by $b$ and $d$ in the PMI-filtered selection, under the assumption that $b^d$ is fixed, and, therefore, can be considered a constant, then the size of the matrixes are polynomial in size in the number of ground atoms the target atom $h$ represents.

**Questions**  $>>$ The lifting process is intricate and heuristic-driven. How sensitive are the final results to these heuristics (e.g., the ordering strategy)? Could you provide a small, concrete example illustrating how the variable assignment rules (1-4) operate across two overlapping windows?

Thanks for pointing out that the description of the lifting process was difficult to follow. We agree, and in the revised version, we will make the procedure clearer in two ways. First, we present the variable-assignment rules in a more algorithmic and structured manner, so that the ordering strategy and the decision points are explicit. Second, we will include a small, concrete example in the appendix (due to space constraints in the main paper) showing how two partially overlapping windows are lifted step-by-step.

$>>$ Could the authors clarify the interval translation process in L354-359? Assuming a window size of 8 (center at 4), how exactly does a learned interval [2, 5] translate into "$\blacksquare$[0,2] $\gamma$ (X,Y) $\wedge$ $\square$[0,1] $\gamma$ (X,Y)"? aka, 2 time units in the past and 1 time unit in the future. The interval relative to the center seems to be [-2, 1].

Yes, but in the syntax of DatalogMTL for each temporal modality (always and eventually), there are two modes: past and future. Therefore, if 0 is the center timepoint, then (past) $\blacksquare$[0,2] means [-2, 1] and (future) $\square$[0,1] means [0, 1]. Their union yields [-2, 1].

$>>$ How does the memory consumption scale with the max_int parameter? Could this become a bottleneck for datasets with complex temporal patterns?

This issue could arise when the window size is very large and many atoms become true within the same window but over multiple disjoint intervals. We did not explicitly discuss this case, as it did not occur in the ICEWS datasets we used. However, such scenarios are easy to imagine, especially in real-world cyber-physical systems. We agree this is an interesting limitation, and we will definitely investigate it in future work.

$>>$ Could you clarify the discrepancy between the abstract's claim of applicability to dense time and the discrete-time semantics defined in Section 2?

Thanks for highlighting this point. To keep the presentation simple, Section 2 focuses on the discrete-time semantics commonly used in prior work. However, the approach itself is not restricted to discrete timelines. By suitably adapting the preprocessing stage— specifically, how temporal intervals are extracted and represented— our framework can operate over dense time as well. We will clarify this distinction in the revised version and we will postpone a clear handling of the dense timeline as a future work.

C.4  Fourth Reviewer

$>>$ No complete guiding example. One piece of example at lines 356–359 is unclear.

Thank you for pointing this out. In the revised version, we provided a more algorithmic presentation of the proposed procedures.

>> Figure 1, which should add substantial information/context to the paper, is underdeveloped.

We agree that Figure 1 can be improved. In the revised version, we updated it to better align with the pipeline steps and provide clearer contextual information.

>> Too many mathematical flaws and presentation inconsistencies in the background and method sections (and beyond). For example:

Many thanks for the careful reading, we fixed the typos.

>> Now for the experiments. This part feels weak and underdeveloped. What are the "sampled variants of MTLearn"?

These are variations of MTLearn that we introduced as the basic version as the basic version of the tool produced a large amount of data we were not able to run either our evaluation script or the MTLearn's one. In order to still present some results, we considered a sampled version of the resulting rules produced by MTLearn.

>> I assume that some examples have been sampled to produce the results in Table 1. If so, have you used different random seeds for the sampling and aggregated the results in Table 1? If not, why?

Yes, we used different random seeds. We will make it clearer in the revised version of the paper.

>> What are ICEWS14 and ICEWS05-15? I assume those are datasets, given the context in Table 1. Do such datasets have a bibliographic reference? What are the characteristics of these datasets?

The specs about ICEWS14 and ICEWS05-15, which are standard benchmarks given that we cited different tools that have been tested against these two datasets in Table 1, are reported in Table 1 from Appendix A.1.

>> I'm asking because the abstract claims the extraction of a linear number of rules (in terms of the dimensionality of the training dataset(s)).

Yes, for each atom we want to learn, we may produce at most a rule, therefore the claim holds.

>> As noted earlier, the cyber-physical scenario requires further explanation (see the Summary).

We will add some details in the appendix, as the main text is already quite full of information.

>> Finally, what is a "strong version of preconditions" (line 450)?

It means that it is not just a precondition for the action, but also a triggering condition, meaning that whenever it holds, the action is executed. We will make it clearer.

>> I've looked at the Appendix, which needs significant improvement, as with the main text. I'm not a huge fan of using bullet points/enumerations for listing just one item (e.g., lines 652–654, line 673, and line 680). It is clear to me that the Appendix hadn't been read before submission. Although notable, the ablation study is weak; I would have expected, besides ablating against eventually or not, to see how other components of the architecture perform while enabling/disabling those parts.

We are happy to extend the ablation studies to analyze additional components of the architecture and would welcome suggestions from the reviewers. Table 11 already examines the effect of removing the eventuality operators, and we will highlight this more clearly in the revised version.

## C.5    Fifth Reviewer

>> In the introduction, how to define a model as "fully differentiable"? It's unclear because this is the core contribution. Many previous works can produce "human-readable rules", so how can to differentiate from those works, and how do fully differentiable architectures generate explicit rules?

This is the first work that produces DatalogMTL rules in a differentiable way. In Section 3, we show how we can express the immediate consequence operator in a neural network.

>> What's the number of parameters used in this model?

The number of parameters depends on the dataset. After generating first-order atoms and filtering them using PMI with the hyperparameters $b$ (breadth) and $d$ (depth), each target atom $h$—which represents $n_r^h$ grounded atoms—has approximately $n_r^h \cdot n^h$ learnable parameters. Here, $n^h \leq 3b^d$, which is intentionally kept small to limit the number of candidate body atoms and ensure tractable model size.

