# OpenReview forum: "Fully Differentiable Temporal FO Rule Learning"
_ICLR.cc/2026/Conference — ICLR 2026 Conference Withdrawn Submission_

### Official Review · Reviewer_sEuH · 2025-10-18

**Soundness:** 2
**Presentation:** 2
**Contribution:** 1
**Rating:** 2
**Confidence:** 3

**Summary:**

This paper proposes a method to learn rules in an end-to-end manner. This paper explores different use cases and demonstrate its benefits.

**Strengths:**

1. Background information and preliminaries are provided to help readers know this domain better.

2. Experimental details are provided.

Overall, this paper is far from reaching ICLR publication standards.

**Weaknesses:**

1. The contribution is, in general, not clear. In the introduction, the core contribution is not clearly delivered (basically one paragraph with several bullet points). The core contribution and methodology are not clear in many points. See questions below for details.

The contribution #1 seems to be weak as it can only learn two specific rules. Also, the "somewhere" statement looks confusing as it does not establish the significance in the given research fields. This looks like a specific technical point without obvious benefits for the overall rule learning framework.

2. Many presentation issues exist. The overall method is not clear to the reviewer. Many methodological concerns exist.

(1) No sub-sections are provided. The overall paper looks pretty dense and hard to digest.

(2) Fig. 1 is pretty unclear. No descriptions are provided, and links to the methodology body are vague.

(3) The reviewer does not find any MLP layers or other common neural network layers. Therefore, it's hard to claim the model as a "neural network". It seems that the proposed method is merely an optimization problem over a set of pre-defined logical operations. It has nothing to do with neural networks.

(4) The preprocessing procedure is very, very confusing. It should be presented with algorithm boxes along with examples. Otherwise, it's hard to follow the dense textual descriptions. Meanwhile, the preprocessing steps do not provide rationales behind, and these model ingredients are not ablated well.

3. The results are much worse than baselines.

**Questions:**

1. In the introduction, how to define a model as "fully differentiable"? It's unclear because this is the core contribution. Many previous works can produce "human-readable rules", so how can to differentiate from those works, and how do fully differentiable architectures generate explicit rules?

2. What's the number of parameters used in this model?

---

> ### Author Response · Authors · 2025-11-26
> **Questions**
>
> $>>$ In the introduction, how to define a model as "fully differentiable"? It's unclear because this is the core contribution. Many previous works can produce "human-readable rules", so how can to differentiate from those works, and how do fully differentiable architectures generate explicit rules?
>
> This is the first work that produces DatalogMTL rules in a differentiable way. In Section 3, we show how we can express the
> immediate consequence operator in a neural network.
>
> $>>$ What's the number of parameters used in this model?
>
> The number of parameters depends on the dataset. After generating
> first-order atoms and filtering them using PMI with the hyperparameters
> \(b\) (breadth) and \(d\) (depth), each target atom \(h\)—which
> represents \(\numberOfRules\) grounded atoms—has approximately
> \(\numberOfRules \cdot \numberOfRelevantAtoms\) learnable parameters.
> Here, \(\numberOfRelevantAtoms \leq 3 b^{d}\), which is intentionally
> kept small to limit the number of candidate body atoms and ensure
> tractable model size.

---

> > ### Comment · Reviewer_sEuH · 2025-11-26
> >
> > Thanks for your rebuttal.
> > The reviewer cannot agree with the point "It's unclear because this is the core contribution."
> >
> > This sentence has a wrong logic. In any academic papers, core contributions have to be clear without diving into technical details.

---

> > > ### Author Response · Authors · 2025-12-04
> > >
> > > Thank you for the comment. We improved the exposition in Section 3 to make the contribution clearer. Our core contribution is enabling the learning of DatalogMTL programs, which in turn allows temporal predictions to be derived through logical inference.

---

### Official Review · Reviewer_KKuL · 2025-10-28

**Soundness:** 1
**Presentation:** 1
**Contribution:** 2
**Rating:** 2
**Confidence:** 4

**Summary:**

The paper presents a differentiable architecture for learning (modal-logic-inspired) temporal first-order rules with metric operators (e.g., properties must hold within a given time interval). A crucial part of the architecture is the use of immediate-consequence operator(s) over data by Walega et al. (2021), which enables the learning of minimal (Herbrand) models using fixpoint semantics. Unfortunately, the paper fails to deliver on many of its promises. From the abstract, the "model [$\ldots$] produces a linear number of rules in the size of the training set", but the paper does not comprehensively analyze this aspect; also, "its seamless applicability to data over both discrete and dense time intervals", but the paper's focus is only on discrete time intervals, while the continuous/dense scenario is only discussed in the last section as future outlook ("our ongoing work aims to support dense timelines [$\ldots$]"). I was genuinely eager of learning more about the method, but the current presentation does not help in this endeavor; specifically, the mathematics are flawed, and the paper's adopted unconventional strategy of not having a guiding example to stress the reader's understanding or to "correct the flawed mathematics" severely hampers the accessibility of this work. Moreover, regarding the experiments, I was unable to fully grasp the setting and the claims; for example, in the cyber-physical scenario, how long the time horizon is, what types of intervals it uses (I assume they were discrete), and how many queries were generated remain a mystery. In summary, even though I have had a strong interest in learning something new, the paper failed to meet my expectations, so I will recommend rejection.

**Strengths:**

(S1) The paper investigates an interesting problem.

(S2) The exploitation of the fixpoint semantics adds foundations to the work.

**Weaknesses:**

(W1) The abstract should be rewritten substantially, as the paper fails to deliver such promises.

(W2) No complete guiding example. One piece of example at lines 356--359 is unclear.

(W3) Figure 1, which should add substantial information/context to the paper, is underdeveloped.

(W4) Too many mathematical flaws and presentation inconsistencies in the background and method sections (and beyond). For example:
  - line 036: The triple $(s,r,o)$ is undefined.
  - lines 124--128: $\mathcal P$ is the set of *ground* atoms with the target predicate/atom, and are called positive examples; one should similarly define $\mathcal N$ (the negative examples). I'm pointing this out because from the current presentation, the target atom $h$ is never used. (*My takeaway:* For context, this is the setting of learning from entailments.)
  - lines 134--139: The head atom $\mathbf v$ is never used/defined. Instead, the presentation uses $\mathbf v_h$. Also, "The vector $\mathbf v_h$ encodes the Boolean values of the body atoms of $r_k$ under $\theta$ [$\ldots$]", but what is the relation between $\mathbf v_h$ and $r_k$? Perhaps it should be $\pi_h$ instead of $r_k$ since $r_k \in \pi_h$.
  - line 177: In $p_{i_j} (e_{k_j}, e_{s_j}) @ t_j$ do you really need all those subscripts? I'm asking because $i$ is never used. Also, $k$ and $s$ (either with subscripted $j$ or not) are not defined.
  - lines 177 & 368: Why not use the same notation for (temporal) facts?
  - lines 191, 304 & 308: Is there any semantical difference between $\times$ and $\cdot$? I would say not.
  - line 199: "[$\ldots$] $(e_i, e_j)$ has the highest co-occurring score [$\ldots$]". What is a co-occurring score at this point of the presentation?
  - line 206: The symbols $m$ and $n$ have been already used previously. Please avoid symbol clashing as much as possible.
  - line 243: What exactly is max_int? Is it the maximum integer that can be stored on a particular hardware? Or is it a constant/hyperparameter that depends on something that I don't see?
  - line 256: I want to stress out the "unhappy" mathematical notation $n^h_r \times 3 n^h$. Is this really the best possible notation that you can come up with? In general, I would advise against these kinds of notations. The entire paper overburdens the reader's cognitive load with subscripts and superscripts.
  - lines 259--260: "[$\ldots$] always-metric atoms [$\ldots$] eventually-metric atoms [$\ldots$]", but always- and eventually-metric atoms have never been defined as such; I mean, these terms are introduced here for the first time. Given my background knowledge, I assume those are the metric atoms that use only (future/past) metric diamonds and boxes, respectively.
  - line 275: "[$\ldots$] using a sigmoid function [$\ldots$]". Which sigmoid function? Does it have a name or definition?
  - line 277: $\mathbf E^{in}$ is undefined.
  - lines 280 & 284: The symbol $\odot$ is undefined. It is, however, defined later at line 284. This, again, is unconventional formal writing.
  - line 286: "[$\ldots$] punctual matrix [$\ldots$]". What is a punctual matrix? As a reader with a PhD in mathematics, I assumed it was $\mathbf P$.
  - lines 273--300: $\mathbf A^i$ for all $i \in \\{1,\ldots, 7\\}$ is an overkill. By the way, at line 283, you use $\mathbf A_3$, while later you use $\mathbf A^3$.
  - line 298: Following the definition, $\mathbf T$'s dimension is $n^h_r \times 3 n^h$ (line 256), and thus $\tilde{\mathbf T}$'s too (line 272). So, by my count, $\tilde{\mathbf T}[:, 0:n^h] || \tilde{\mathbf T}[:, n^h:2n^h] || \tilde{\mathbf T}[:, 2n^h:3n^h]$, where $||$ is the matrix concatenation operator, has dimension $n^h_r \times (3n^h + 3)$. Therefore, the definition of $\mathbf B$ is wrong. Moreover, again, $\mathbf T^{in}$ is undefined.
  - line 316: The paper uses $p^k$ for the penalty. There are at least two issues here: (1) $p$ has already been used for the atoms, and (2) $k$ has already been used as well.
  - line 349: Both $\mathbf S_{i,j}$ and $\mathbf E_{i,j}$ are undefined.
  - lines 360--364: $D,t \models H$ is undefined. Moreover, the equation's definition of $w(h,r)$ has at least three issues: (i) $h$ is never used in the right-hand part of the equation, (ii) $sc(\cdot)$ is undefined, and (iii) $hc(\cdot)$ is undefined as well.
  - line 370: Once more, $c_{max}$ is undefined.
  - line 410: $r[D]$ is undefined.
  - line 419: "[$\ldots$] $R(a,b)@t$, where $t_{max}$ is the maximum timestamp in $D$ [$\ldots$]". What is the role of $t_{max}$ here?

As can be noted, the above (non-comprehensive) list of inconsistencies and flaws severely detracts from the overall understandability of the core proposal. And again, I was genuinely eager to understand the proposal, but even with that in mind, I was unable to put the pieces together. As a general rule, I would advise doing a thorough proofread before submission.

(W5) Now for the experiments. This part feels weak and underdeveloped. What are the "sampled variants of MTLearn"? I assume that some examples have been sampled to produce the results in Table 1. If so, have you used different random seeds for the sampling and aggregated the results in Table 1? If not, why? What are ICEWS14 and ICEWS05-15? I assume those are datasets, given the context in Table 1. Do such datasets have a bibliographic reference? What are the characteristics of these datasets? I'm asking because the abstract claims the extraction of a linear number of rules (in terms of the dimensionality of the training dataset(s)). As noted earlier, the cyber-physical scenario requires further explanation (see the Summary). Finally, what is a "strong version of preconditions" (line 450)?

(W6) I've looked at the Appendix, which needs significant improvement, as with the main text. I'm not a huge fan of using bullet points/enumerations for listing just one item (e.g., lines 652--654, line 673, and line 680). It is clear to me that the Appendix hadn't been read before submission. Although notable, the ablation study is weak; I would have expected, besides ablating against eventually or not, to see how other components of the architecture perform while enabling/disabling those parts.

Minors:
- line 456 uses $:-$ while it should be $\leftarrow$
- Please fix the naming inconsistency of MTLearn vs MT-Learn (line 468).

**Questions:**

See above.

---

> ### Author Response · Authors · 2025-11-26
> **Questions**
>
> $>>$ No complete guiding example. One piece of example at lines 356--359 is unclear.
>
> Thank you for pointing this out. In the revised version, we provide a
> more algorithmic presentation of the proposed procedures. If space
> allows, we will also integrate a concise running example to make the
> workflow clearer.
>
> $>>$ Figure 1, which should add substantial information/context to the paper, is underdeveloped.
>
> We agree that Figure 1 can be improved. In the revised version, we
> will update it to better align with the pipeline steps and provide
> clearer contextual information.
>
> $>>$ Too many mathematical flaws and presentation inconsistencies in the background and method sections (and beyond). For example:
>
> Many thanks for the careful reading, we will fix the typos.
>
>
> $>>$ Now for the experiments. This part feels weak and underdeveloped. What are the "sampled variants of MTLearn"?
>
> These are variations of MTLearn that we introduced as the basic version
> of the tool was not runnable.
>
>
> $>>$ I assume that some examples have been sampled to produce the results in Table 1. If so, have you used different random seeds for the sampling and aggregated the results in Table 1? If not, why?
>
> Yes, we used different random seeds. We will make it clearer in the
> revised version of the paper.
>
> $>>$ What are ICEWS14 and ICEWS05-15? I assume those are datasets, given the context in Table 1. Do such datasets have a bibliographic reference? What are the characteristics of these datasets?
>
> The specs about ICEWS14 and ICEWS05-15, which are standard benchmarks
> given that we cited different tools that have been tested against these
> two datasets in Table 1, are reported in Table 1 from Appendix A.1.
>
> $>>$ I'm asking because the abstract claims the extraction of a linear number of rules (in terms of the dimensionality of the training dataset(s)).
>
> Yes, for each atom we want to learn, we may produce at most a rule,
> therefore the claim holds.
>
> $>>$ As noted earlier, the cyber-physical scenario requires further explanation (see the Summary).
>
> We will add some details in the appendix, as the main text is already
> quite full of information.
>
>
> $>>$ Finally, what is a "strong version of preconditions" (line 450)?
>
> It means that it is not just a precondition for the action, but also
> a triggering condition, meaning that whenever it holds, the action
> is executed. We will make it clearer.
>
> $>>$ I've looked at the Appendix, which needs significant improvement, as with the main text. I'm not a huge fan of using bullet points/enumerations for listing just one item (e.g., lines 652--654, line 673, and line 680). It is clear to me that the Appendix hadn't been read before submission. Although notable, the ablation study is weak; I would have expected, besides ablating against eventually or not, to see how other components of the architecture perform while enabling/disabling those parts.
>
> We are happy to extend the ablation studies to analyze additional
> components of the architecture and would welcome suggestions from the
> reviewers. Table 11 already examines the effect of removing the
> eventuality operators, and we will highlight this more clearly in the
> revised version.

---

### Official Review · Reviewer_AwN9 · 2025-10-28

**Soundness:** 3
**Presentation:** 3
**Contribution:** 3
**Rating:** 6
**Confidence:** 2

**Summary:**

This paper introduces MT-Diff-Learn, a novel, fully differentiable architecture for learning first-order temporal logic rules expressed in DatalogMTL. DatalogMTL extends Datalog with metric temporal operators (e.g., "always", "eventually") annotated with explicit time intervals. The framework aims to address limitations in existing temporal rule learning methods, specifically their limited expressiveness and scalability issues often caused by massive rule generation or the need to reify timestamps. The methodology involves a pipeline: data preparation (sliding windows, heuristic-based lifting of ground facts, PMI filtering), a neural network that learns weights and interval boundaries using differentiable approximations of interval containment (for $\Box$) and overlap (for $\diamond$), and a final rule extraction phase.

**Strengths:**

•	MT-Diff-Learn is a fully differentiable framework for learning DatalogMTL rules. The inclusion of mechanisms to learn eventuality conditions ($\diamond$, $\bullet$) significantly increases the expressiveness compared to prior temporal rule learning methods, allowing the model to capture temporal uncertainty.

•	The approach demonstrates excellent scalability by generating several rules linear in the size of the training set. Empirically, it produces orders of magnitude fewer rules than MTLearn. This vastly improves interpretability.

•	By treating intervals as first-class citizens and learning their bounds directly, the model avoids the quadratic blow-up associated with reifying every possible timestamp, a common bottleneck in other approaches.

•	The synthetic scenarios (Cyber-Physical and Action Description) effectively demonstrate the utility of the eventuality operators. The ablation study confirms their importance, showing a sharp performance drop when they are disabled.

**Weaknesses:**

•	The lifting process is complex and relies heavily on heuristics (overlap scores, processing order, variable assignment constraints). The robustness of the system to these choices is unclear, and the impact of associated hyperparameters (window size $l$, breadth $b$, depth $d$) is not analyzed. The generalizability of this lifting process warrants further investigation.

•	On standard tKG benchmarks (ICEWS), MT-Diff-Learn significantly lags behind SOTA embedding-based methods. While the focus is on interpretability, this gap may limit adoption where accuracy is paramount. The paper should better discuss this trade-off.

•	Key parts of the methodology are difficult to parse. The lifting procedure (W1) is confusing. Furthermore, the process of translating the learned intervals back into DatalogMTL syntax is unclear. The example provided (transforming [2, 5] into a conjunction of past and future operators) seems overly complicated and requires clarification regarding how semantics are preserved relative to the window center.

•	The abstract claims applicability to data over dense time intervals. However, the semantics defined in Section 2 are explicitly over integers ($\mathbb{Z}$), and dense time is only mentioned as future work.

•	While the output model is succinct, the training process involves complex tensor operations. The input tensors (Start/End) have a dimension representing the maximum number of disjoint intervals an atom might hold in a window. The paper does not detail the practical implications of this on memory usage and training time if this value is large.

**Questions:**

The lifting process is intricate and heuristic-driven. How sensitive are the final results to these heuristics (e.g., the ordering strategy)? Could you provide a small, concrete example illustrating how the variable assignment rules (1-4) operate across two overlapping windows?

Could the authors clarify the interval translation process in L354-359? Assuming a window size of 8 (center at 4), how exactly does a learned interval [2, 5] translate into "■[0,2] \gamma (X,Y) ∧ □[0,1] \gamma (X,Y)"? aka, 2 time units in the past and 1 time unit in the future. The interval relative to the center seems to be [-2, 1].

How does the memory consumption scale with the max_int parameter? Could this become a bottleneck for datasets with complex temporal patterns?

Could you clarify the discrepancy between the abstract's claim of applicability to dense time and the discrete-time semantics defined in Section 2?

---

> ### Author Response · Authors · 2025-11-26
> **Weaknesses**
>
> \paragraph{Weaknesses}
>
> $>>$ The lifting process is complex and relies heavily on heuristics (overlap scores,
> processing order, variable assignment constraints). The robustness of the system
> to these choices is unclear, and the impact of associated hyperparameters
> (window size $l$, breadth $b$, depth $d$) is not analyzed. The generalizability
> of this lifting process warrants further investigation.
>
> We agree that more ablation studies should be provided, and we aim to integrate them in the revised version of the paper.
>
> $>>$ On standard tKG benchmarks (ICEWS), MT-Diff-Learn significantly lags behind SOTA embedding-based methods. While the focus is on interpretability, this gap may limit adoption where accuracy is paramount. The paper should better discuss this trade-off.
>
> We agree that embedding-based temporal KGC methods achieve higher raw accuracy
> on large-scale benchmarks such as ICEWS. However, these models are inherently
> black-box and do not yield interpretable temporal rules. We will clarify this
> trade-off in the revised version and position our approach as an interpretable
> alternative rather than a direct competitor to high-performance embedding models.
> Furthermore, in cases where the labels of entities and relation names do not
> carry any semantic information, the performance of embedding-based approaches
> may degrade significantly, whereas rule-based methods remain robust to such
> semantic neutrality.
>
> $>>$ Key parts of the methodology are difficult to parse. The lifting procedure (W1) is confusing. Furthermore, the process of translating the learned intervals back into DatalogMTL syntax is unclear. The example provided (transforming [2, 5] into a conjunction of past and future operators) seems overly complicated and requires clarification regarding how semantics are preserved relative to the window center.
>
> The translation from learned intervals into DatalogMTL is conceptually simple, so that part should be made more readable if it causes any problem.
>
> $>>$ The abstract claims applicability to data over dense time intervals. However, the semantics defined in Section 2 are explicitly over integers ($\mathbb{Z}$), and dense time is only mentioned as future work.
>
> Yes, with some adaptation in the preprocessing of the data we can
> apply this method to learn also rules operating on a dense timeline,
> however, to ease the presentation we stick to the simpler case of
> the integers. We will make this clearer in the revised version.
>
> $>>$ While the output model is succinct, the training process involves complex tensor operations. The input tensors (Start/End) have a dimension representing the maximum number of disjoint intervals an atom might hold in a window. The paper does not detail the practical implications of this on memory usage and training time if this value is large.
>
> For each target atom $h$, we generate the three tensors for (i)
> the truth matrix, (ii) the starting and (iii) the ending matrix.
> Since the number of the relevant atoms is parametrized by $b$ and $d$
> in the PMI-filtered selection, under the assumption that $b^d$ is
> fixed, and, therefore, can be considered a constant, then the size of
> the matrixes are polynomial in size in the number of ground atoms the
> target atom $h$ represents.

---

> ### Author Response · Authors · 2025-11-26
> **Questions**
>
> \paragraph{Questions}
>
> $>>$ The lifting process is intricate and heuristic-driven. How sensitive are the final results to these heuristics (e.g., the ordering strategy)? Could you provide a small, concrete example illustrating how the variable assignment rules (1-4) operate across two overlapping windows?
>
> Thanks for pointing out that the description of the lifting process
> was difficult to follow. We agree, and in the revised version, we will
> make the procedure clearer in two ways. First, we present the
> variable-assignment rules in a more algorithmic and structured manner,
> so that the ordering strategy and the decision points are explicit.
> Second, we will include a small, concrete example in the appendix (due
> to space constraints in the main paper) showing how two partially
> overlapping windows are lifted step-by-step.
>
> $>>$ Could the authors clarify the interval translation process in L354-359? Assuming a window size of 8 (center at 4), how exactly does a learned interval [2, 5] translate into "$\blacksquare$[0,2] $\gamma$
> (X,Y) $\land$
> $\square$[0,1] $\gamma$ (X,Y)"? aka, 2 time units in the past and 1 time unit in the future. The interval relative to the center seems to be [-2, 1].
>
> Yes, but in the syntax of DatalogMTL for each temporal modality
> (always and eventually), there are two modes: past and future.
> Therefore, if 0 is the center timepoint, then (past) $\blacksquare$[0,2] means [-2, 1] and (future) $\square$[0,1] means [0, 1]. Their union yields [-2, 1].
>
> $>>$ How does the memory consumption scale with the max\_int parameter? Could this become a bottleneck for datasets with complex temporal patterns?
>
> This issue could arise when the window size is very large and many
> atoms become true within the same window but over multiple disjoint
> intervals. We did not explicitly discuss this case, as it did not
> occur in the ICEWS datasets we used. However, such scenarios are easy
> to imagine, especially in real-world cyber-physical systems. We agree
> this is an interesting limitation, and we will definitely investigate
> it in future work.
>
> $>>$ Could you clarify the discrepancy between the abstract's claim of applicability to dense time and the discrete-time semantics defined in Section 2?
>
> Thanks for highlighting this point. To keep the presentation simple,
> Section 2 focuses on the discrete-time semantics commonly used in
> prior work. However, the approach itself is not restricted to discrete
> timelines. By suitably adapting the preprocessing stage---
> specifically, how temporal intervals are extracted and represented---
> our framework can operate over dense time as well.
> We will clarify this distinction in the revised version and we will
> postpone a clear handling of the dense timeline as a future work.

---

### Official Review · Reviewer_mtGT · 2025-10-29

**Soundness:** 3
**Presentation:** 2
**Contribution:** 3
**Rating:** 6
**Confidence:** 2

**Summary:**

The paper presents MT-Diff-Learn, a fully differentiable neural architecture for learning First-Order Temporal Logic (FOTL) rules enhanced with metric operators. The model extends differentiable immediate consequence operators, common in differentiable logic programming, to the temporal setting. This extension enables the system to simultaneously learn predicates, the logical structure of rules, and the associated precise metric temporal intervals. The resulting approach is claimed to handle data over both discrete and dense time, capturing temporal dependencies efficiently without reifying all timestamps. The authors state that this leads to a rule complexity that scales linearly with the training data size, positioning the model as a mechanism for discovering complex, interpretable temporal rules.

**Strengths:**

The primary strength of this work is the integration of metric temporal operators within a fully differentiable rule learning framework. This capability is notable, as it allows the approach to discover rules containing complex temporal patterns over dense time, addressing a limitation of methods restricted to discrete time steps or simple precedence relations. The ablation study effectively isolates the contribution of the eventuality operators, demonstrating a substantial performance gain (e.g., in MRR), which supports the model's design for capturing system dynamics where events occur with uncertainty within a temporal window. Furthermore, the claim of generating a linear number of rules relative to the training data size suggests beneficial characteristics for scalability and model complexity in applications involving large, complex datasets.

**Weaknesses:**

The experimental section could be strengthened by including comparisons against more diverse state-of-the-art neuro-symbolic or temporal rule mining baseline approaches, beyond standard non-temporal or limited temporal inductive logic programming methods, to fully contextualize the proposed model's performance in the broader field of temporal sequence modeling.

**Questions:**

How robust is the learning of the continuous metric interval bounds to noise in the training data, and what regularization or loss terms (if any) are specifically implemented to prevent interval collapse or explosion during gradient descent?

Given the focus on interpretability, can the authors provide a more detailed analysis of the learned rules—perhaps a qualitative summary or examples from the different use cases—to illustrate how MT-Diff-Learn discovers non-obvious or complex temporal relationships that purely sequential models might miss?

---

> ### Author Response · Authors · 2025-11-26
> **Weaknesses**
>
> \paragraph{Weaknesses}
>
> $>>$ The experimental section could be strengthened by including comparisons against more diverse state-of-the-art neuro-symbolic or temporal rule mining baseline approaches, beyond standard non-temporal or limited temporal inductive logic programming methods, to fully contextualize the proposed model's performance in the broader field of temporal sequence modeling.
>
>
> We interpret the reviewer's comment as referring to temporal sequence modeling.
> Our focus, however, is on interpretable rule learning, not black-box sequence
> modeling. While sequence models such as LSTMs or Transformers can handle
> temporal prediction, they do not produce symbolic rules or interpretable
> structures, making them unsuitable baselines for our setting. Of course, the
> literature is very huge, so we reported only some results of temporal sequence
> modeling tools in the Table about
> temporal link prediction in the interpolation settings.
> Nonetheless, we will clarify this distinction in the paper.

---

> ### Author Response · Authors · 2025-11-26
> **Questions**
>
> \paragraph{Questions}
>
> $>>$ How robust is the learning of the continuous metric interval bounds to noise in the training data,
>
> Our syntactic dataset is obtained by adding some uncertainty on the realization of the grant via the parameters $\mu$ and $\sigma$, and some noise is injected as the grant is given if a uniform probability variable that uniformly ranges from 0 to 1 exceeds .9.
>
> $>>$ and what regularization or loss terms (if any) are specifically implemented to prevent interval collapse or explosion during gradient descent?
>
> We have not considered such regularization.
>
> $>>$ Given the focus on interpretability, can the authors provide a more detailed analysis of the learned rules—perhaps a qualitative summary or examples from the different use cases—to illustrate how MT-Diff-Learn discovers non-obvious or complex temporal relationships that purely sequential models might miss?
>
> An example of a rule that is learned in the ICEWS setting is:
> \[
> consult(X, Y) \leftarrow \Box_{[0, 0]} consult(Y, X).
> \]
> This rule is not di per see involving a temporal complex structure, but it showcase how certain property such as symmetry in this case can be easily
> represented.

---

### Official Review · Reviewer_Lixg · 2025-11-04

**Soundness:** 2
**Presentation:** 1
**Contribution:** 2
**Rating:** 2
**Confidence:** 2

**Summary:**

This paper is a technical contribution within a dense existing project.

It builds on [Gao et al. (2024)](https://www.sciencedirect.com/science/article/pii/S0004370224000444).  That paper is not easy to skim.  Its [2022 version](https://www.ijcai.org/proceedings/2022/417) is not much easier, but it is at least shorter and has a video that sort of walks through an example (see also the [appendices](https://arxiv.org/pdf/2204.13570)).  My *rough* understanding of that prior work after wrestling with it for a while is as follows (the authors can correct me if I'm wrong):
* The goal is to find a logic program that predicts the truth values of some given positive and negative atoms given a set of background fact atoms.
* This is done by optimizing a soft Datalog program and then somehow extracting a hard Datalog program from it.
* A soft program is similar to a hard program except that it replaces AND, OR with fuzzy differentiable versions.
* The learning objective for the soft program is the log-loss on the fuzzy predictions, plus several regularizers.
* Crucially, the set of non-ground terms that can appear in the program is bounded in advance since the maximum number of predicates and variables are given as hyperparameters, the arity of each predicate is always 1 or 2, and the program contains no constants.
* For *each* non-ground term, a matrix is learned whose rows correspond to the rules with that non-ground term as head.  Each row has entries in [0,1] and specify the degree to which each of the non-ground terms participate in the fuzzy AND for that rule's body.  (The number of rows is a hyperparameter, and there is one column for each non-ground term.)

**The present paper extends that method to [DatalogMTL](https://arxiv.org/abs/2201.04596),** or rather continues an extension already begun by [MTLearn](https://proceedings.kr.org/2024/90/kr2024-0090-wang-et-al.pdf).

DatalogMTL is a version of Datalog in which each predicate implicitly has an additional argument Time, so that it can be true at some times and not others.  A rule essentially has the form property(Time, Arg1, Arg2) :- ..., where each subgoal has the form (∀T ∈ [Time+a,Time+b]) property(T,X,Y) or (∃T ∈ [Time+a,Time+b]) property(T,X,Y).  These additional levels of quantification will complicate learning.

In the learning method here, the provided positive facts are fully ground, including the Time argument.  However, in contrast to the prior work, there are no negative facts or background facts provided.  (At least, this is what I conclude from Fig. 1.)  Thus, the goal may be to compress the provided set of positive facts by writing a short program that predicts some of them from others. That would make it a kind of association rule mining method (traditional example: "if you buy beer, you're also likely to buy chips").

**Strengths:**

The authors appear to have worked hard.  I think they themselves probably understand what they're doing, even if it is hard for a reader to understand from this presentation.

**Weaknesses:**

I apologize that I was not able to follow the paper well.  (I thought it would help to go back to the prior work, but it didn't.)

The paper is a bit like reading someone's half-documented code.  The paper says "We do this, then we do that, then we do that."  But it is often not clear what the goal is or why those steps are appropriate.  This is particularly a problem in section 3.  It also makes it difficult to adjust for lapses in the exposition.  To take some early examples: "target atom $h$" isn't defined at line 126 and it's not clear where it comes from or why it's needed.  At line 132, it's not clear where the threshold comes from: before relaxation, is it meant to be the number of negative literals in the rule?  At line 202, what happens if *multiple* atoms $p(X_i,X_j)$ were "already considered in the previous windows"?

Even as a low-level description of what's done, the paper doesn't quite hang together.  To take one example: Line 190 defines the "overlap score" $O(w)$ and says that it is "used to determine processing order," but it's not clear what "processing order" refers to, and neither $O(w)$ nor the overlap score is ever mentioned again!  (I suppose it's the same as the "co-occurring score" in the next paragraph.)  Furthermore, the definition of $O(w)$ contains unbound variables $i,j$.  (I suppose it should have been named $O_{i,j}(w)$.)

**Questions:**

What is the formal learning problem here?
* Is it a machine learning problem where there is a true set of temporal facts over intervals, the answers to some queries are observed at random, and the estimated DatalogMTL program should accurately predict the answers to other queries?
* Is it a statistics problem where a true parameter is a DatalogMTL program, the answers to some queries are observed at random, and the true program should be recovered in the limit?  If so, is the program in fact identifiable?
* Is it a computational problem of finding a small description of a set of positive facts (and if so, how is it ensured that the description does not predict false facts, since negative examples are not provided)?
* Is it an association rule mining problem as mentioned earlier?  If so, what is the success criterion?

What does "consecutive occurrences" mean at line 181?  You talk about @t, @t+1, and @t+2.  I had assumed that time was continuous, since you referred early on to "intervals"; did you actually intend for time to be discrete?

It's not clear to me whether learning temporal logic programs is an important problem.  Can you make a case for it?  Line 417 discusses a "temporal link prediction" task, but of course there are many cleaner ways to solve such a task, such as training a Transformer with temporal positional embeddings.  (And based on Table 1, it looks like your method wasn't competitive with such alternatives.  You suggest it might be more explainable, but it's hard to evaluate the quality of the explanation from the paper, especially given its size: 40K rules according to line 425.)  Lines 432 and 449 seem to discuss temporal modeling tasks, but these could be handled by generative methods such as the neural Hawkes process or its variants.  (Here maybe you do have an explainability advantage, with 80 rules at line 460 on this synthetic dataset.  But as an alternative, perhaps one could train a neural generative model and then extract explanatory patterns from it post hoc.)

I am not able to follow the details of the method as presented.  On first principles, I would have expected a method similar to Gao et al. (2024), which you present as your starting point, but where the matrix columns had names like (∀T ∈ [Time+a,Time+b]) property(T,X,Y) or (∃T ∈ [Time+a,Time+b]) property(T,X,Y).  These truth conditions of such a column would be softened, in part by fuzzing the edges of the interval [Time+a,Time+b].  Thus, you could improve a, b by following their gradient.  Why didn't you do it this way?

Instead, you used some heuristics involving windows and PMI to find rules.  What happens if you make the window size too large or too small?  Does this change the number of rules you find and their specificity, so that you might underpredict or overpredict positive facts?  What are the simplest examples where your heuristics would fail?

---

> ### Author Response · Authors · 2025-11-26
> **Weaknesses**
>
> \paragraph{Weaknesses}
>
> $>>$ The paper is a bit like reading someone's half-documented code. The paper says ``We do this, then we do that, then we do that.'' But it is often not clear what the goal is or why those steps are appropriate. This is particularly a problem in section 3. It also makes it difficult to adjust for lapses in the exposition. To take some early examples: ``target atom'' isn't defined at line 126 and it's not clear where it comes from or why it's needed.
>
>
> Thanks for your comments. We noticed this issue and updated the presentation in a more schematic style. Please check Section 3 in the revised version.
>
> Regarding the conceptual clarifications: our objective is to learn a logic program $P$ that derives a designated \emph{target atom}~$h$, given background knowledge $B$ and sets $\mathcal{P}$ and $\mathcal{N}$ of positive and negative examples. A valid solution $P$ must entail all positive examples while excluding all negative ones:
> \[
> B \cup P \models e^+ \;\; \forall e^+ \in \mathcal{P},
> \qquad
> B \cup P \not\models e^- \;\; \forall e^- \in \mathcal{N}.
> \]
>
> Instead of relying on a black-box neural predictor, the network is structured so as to learn a program $P$ whose inference behaviour follows the formal semantics of DatalogMTL. Predictions are framed as logical entailment: the model determines whether the target atom (i.e., the output label~$y$) holds.
>
> For example, if \textit{wet\_ground} is the target atom, the system may learn a rule such as:
> \[
> \mathit{wet\_ground} \;\leftarrow\; \blacklozenge[2,0]\,\mathit{rains},
> \]
> which, under the semantics of DatalogMTL, states that the ground is wet if it has rained at any point within the last two days or on the current day in a scenario where the time unit is a day. In general, \emph{target atoms} are precisely those atoms whose truth values the model aims to predict via the learned logical rules.
>
>
> $>>$ At line 132, it's not clear where the threshold comes from: before relaxation, is it meant to be the number of negative literals in the rule?
>
> We wrote: ``Let $v$ be an real number, and let $\varphi(v)= v'$ be the threshold function, where $v' = 1$ if $v \geq \tau$, and $v'_i = 0$ otherwise with $\tau$ being the threshold value.'' Instead of $v'_i = 0$, it should have been $v' = 0$. Perhaps the typo hindered the reading. Therefore, it is not the number of negative literals. It is a threshold value such that if a value is greater than $\tau$, then it collapses to 1, otherwise to 0.
>
> $>>$ At line 202, what happens if multiple atoms were ``already considered in the previous windows''?
>
> If there is more than one candidate, one is randomly picked.
>
> $>>$ Even as a low-level description of what's done, the paper doesn't quite hang together. To take one example: Line 190 defines the "overlap score"
> and says that it is "used to determine processing order," but it's not clear what "processing order" refers to, and neither nor the overlap score is ever mentioned again! (I suppose it's the same as the "co-occurring score" in the next paragraph.)
>
> Thank you for pointing this out. Our heuristics compute two different scores:
> (i) an overlap score used to determine the order in which windows are processed, and (ii) a co-occurrence score used to decide the order in which ground atoms are lifted to first-order ones.
> We intended the textual description to be easier to follow than presenting a full procedural algorithm, but we see how this may have caused confusion. If allowed, we will include a clear algorithmic description in the appendix to make the distinction explicit.
>
>
> $>>$ Furthermore, the definition of contains unbound variables. (I suppose it should have been named .)
>
> Thank you for highlighting this point. As stated in the paper, the number of rules we generate is linear in the input, which implies that the number of variables is indeed bounded. We made it clearer by
> exposing the procedure in an algorithmic fashion: the generation of
> the variables is described from line 14 to line 17 in Algorithm 1.
> One cannot generate more $|E||\mathcal{W}|$ variables, where $E$ denotes the set of all entities and $|\mathcal{W}|$ the number of windows.

---

> ### Author Response · Authors · 2025-11-26
> **Questions**
>
> \paragraph{Questions}
>
> $>>$ What is the formal learning problem here?
>
> The formal learning problem we address is that of learning DatalogMTL rules (defined in Section 2) that describe a temporal dataset. More precisely, our setting corresponds to the temporal interpolation problem studied in prior work such as [1] and [2]: given temporal sequences of facts, the goal is to induce a rule set whose temporal consequences reconstruct or generalize these sequences.
>
>
> [1] Zhen Han, Peng Chen, Yunpu Ma, and Volker Tresp. Explainable subgraph reasoning for forecasting
> on temporal knowledge graphs. In International conference on learning representations, 2020.
>
> [2] Yushan Liu, Yunpu Ma, Marcel Hildebrandt, Mitchell Joblin, and Volker Tresp. Tlogic: Temporal
> logical rules for explainable link forecasting on temporal knowledge graphs. In Proceedings of
> the 36th AAAI Conference on Artificial Intelligence (AAAI 2022), pp. 4120–4127. AAAI Press,
> 2022.
>
> $>>$ Is it a machine learning problem where there is a true set of temporal facts over intervals, the answers to some queries are observed at random, and the estimated DatalogMTL program should accurately predict the answers to other queries?
>
> The learning problems we consider are indeed machine-learning tasks over temporal data, but they span three concrete settings rather than a single abstract formulation:
>
> 1. Temporal link prediction (interpolation) on temporal knowledge graphs, following the setting of [1] and [2], where the goal is to reconstruct unseen temporal facts from partially observed sequences.
>
> 2. A synthetic temporal dataset in which the model must learn temporal relations connecting requests and grants.
>
> 3. A temporal action-learning scenario, where the system induces action descriptions from observations of an autonomous agent.
>
> More generally, we learn a DatalogMTL program that generalizes from observed temporal facts to unobserved ones. We tested it across different temporal reasoning domains to show its versatility.
>
> $>>$ Is it a statistics problem where a true parameter is a DatalogMTL program, the answers to some queries are observed at random, and the true program should be recovered in the limit? If so, is the program in fact identifiable?
>
> No, we did not consider this setting.
>
> $>>$ Is it a computational problem of finding a small description of a set of positive facts
>
> It does not necessarily need to be small, but if it is small, it mean that the rules are more significant and it is easier to make inferences.
>
> $>>$ (and if so, how is it ensured that the description does not predict false facts, since negative examples are not provided)?
>
> In the [1] and [2] settings, an answer is always expected.
>
> $>>$ Is it an association rule mining problem as mentioned earlier? If so, what is the success criterion?
>
> The success criterion is not based on association-rule mining but on standard evaluation metrics used in temporal link prediction and temporal reasoning tasks. In all experiments, we assess performance using Hits@Rank and MRR, which measure how well the learned DatalogMTL program predicts the correct temporal facts among all candidate answers.
>
> $>>$ What does "consecutive occurrences" mean at line 181? You talk about \@t, \@t+1, and \@t+2. I had assumed that time was continuous, since you referred early on to "intervals"; did you actually intend for time to be discrete?
>
> Yes, in our experimental setting time is discrete. This allows us to refer to consecutive occurrences such as
> \@$t$, \@$t+1$, and \@$t+2$, which correspond to the discrete interval $[t,t+2]$.
> Intervals can still be defined naturally over a discrete timeline. Extending the framework to continuous time is an interesting direction for future work; it would mainly require adjustments in data preprocessing rather than changes to the core method.

---

> ### Author Response · Authors · 2025-11-26
> **Questions (2)**
>
> $>>$ It's not clear to me whether learning temporal logic programs is an important problem. Can you make a case for it?
>
> Learning temporal logic programs is important because many real-world reasoning tasks are inherently temporal: events have durations, actions have delayed effects, and relations evolve over time. The ability to induce temporal rules from data places our work within a growing line of research in Inductive Logic Programming (ILP) and neuro-symbolic reasoning, where temporal extensions are becoming increasingly relevant.
>
>
> Temporal logic programs provide high-level, human-readable rules that capture
> how facts evolve over time. These rules are valuable not only for prediction
> (e.g., temporal link prediction, learning action descriptions) but also for
> explainability and diagnosis in evolving systems, where extracting structured
> knowledge from data is crucial. In contrast, transformer-based models—despite
> recent efforts to improve their interpretability, remain inherently more opaque
> than symbolic formalisms, which offer explicit reasoning steps. Symbolic
> approaches may sometimes trade off raw performance, but they provide semantic
> clarity that is essential in domains where understanding temporal dynamics
> matters.
>
> We will make this motivation clearer in the revised version.
>
> $>>$ But as an alternative, perhaps one could train a neural generative model and then extract explanatory patterns from it post hoc.)
>
> Training a neural generative model and extracting patterns post hoc
> is indeed a possible alternative.
> However, our goal is to extract rules that correspond directly to the actual
> decision-making process of the model—i.e., explicit if–then clauses that the
> system truly relies on during inference. Post-hoc pattern results in a knowledge
> that does not offer the same properties of a formal language with a clear
> syntax and semantics. Therefore it cannot directly support classical
> knowledge-representation tasks such as logical inference, automated planning,
> verification, or diagnosis.
> In contrast, our method learns rules within a formal logic framework
> (DatalogMTL), ensuring that the extracted rules can be given as an input to
> temporal rule solvers or monitoring frameworks.
>
> To integrate: ``post-hoc interpretation, we take an existing machine learning system, that has already been trained, and try to
> understand its inner state. In the other approach, designing explicit already-interpretable machine learning systems, we con-
> strain the design of the machine learning system to guarantee, in advance, that its results will be interpretable''
>
> $>>$ I am not able to follow the details of the method as presented. On first principles, I would have expected a method similar to Gao et al. (2024), which you present as your starting point, but where the matrix columns had names like (for all T in [Time+a,Time+b]) property(T,X,Y) or (exists T in [Time+a,Time+b]) property(T,X,Y). These truth conditions of such a column would be softened, in part by fuzzing the edges of the interval [Time+a,Time+b]. Thus, you could improve a, b by following their gradient. Why didn't you do it this way?
>
> We may be misunderstanding the reviewer’s suggestion, but our architecture does
> already allow for differentiable learning of interval endpoints. In DatalogMTL,
> temporal operators such as boxes or diamonds are implemented through
> differentiable layers, and the parameters are learned via gradient descent—
> analogous in the same spirit the reviewer suggests.
>
> $>>$  What happens if you make the window size too large or too small?
>
> The size of the windows is optimized via a grid-search. The idea is that
> the windows should be large enough to capture most of the relevant temporal
> dependencies across the data.
>
> $>>$ Does this change the number of rules you find and their specificity, so that you might underpredict or overpredict positive facts?
>
> Changing the window size can indeed influence the specificity of the learned
> rules. Intuitively, larger windows rely less on temporally local dependencies
> and may therefore lead to more general rules, which can increase the risk
> of overgeneralization.
> On this other side, when the temporal dependencies in the data are short-term,
> increasing the window size may introduce unnecessary complexity and this may
> negatively affect the quality of the learned rules.
> We will include this discussion in the revised version to clarify how window
> size interacts with rule specificity and prediction accuracy with some ablation
> studies to support this.
>
> $>>$ What are the simplest examples where your heuristics would fail?
>
> The simplest example is when every entity appears only once, and the
> first-order ``representatives'' do not generalize. In fact,
> in this case, the
> outcome will be a grounded DatalogMTL program.

---

### Note · Authors · 2026-01-13

I have read and agree with the venue's withdrawal policy on behalf of myself and my co-authors.